# Parkin regulates adiposity by coordinating mitophagy with mitochondrial biogenesis in white adipocytes

Timothy M. Moore [1,13], Lijing Cheng[2,13], Dane M. Wolf[3,10,11], Jennifer Ngo[3], Mayuko Segawa[3,10], Xiaopeng Zhu[4,12], Alexander R. Strumwasser[3], Yang Cao[1], Bethan L. Clifford[1], Alice Ma[3], Philip Scumpia [2], Orian S. Shirihai [3], Thomas Q. de Aguiar Vallim[1,5,6], Markku Laakso[7], Aldons J. Lusis [1,8], Andrea L. Hevener[3,9] & Zhenqi Zhou [3,6] ✉

Parkin, an E3 ubiquitin ligase, plays an essential role in mitochondrial quality control. However, the mechanisms by which Parkin connects mitochondrial homeostasis with cellular metabolism in adipose tissue remain unclear. Here, we demonstrate that *Park2* gene (encodes Parkin) deletion specifically from adipose tissue protects mice against high-fat diet and aging-induced obesity. Despite a mild reduction in mitophagy, mitochondrial DNA content and mitochondrial function are increased in *Park2* deficient white adipocytes. Moreover, *Park2* gene deletion elevates mitochondrial biogenesis by increasing Pgc1α protein stability through mitochondrial superoxide-activated NAD(P)H quinone dehydrogenase 1 (Nqo1). Both in vitro and in vivo studies show that Nqo1 overexpression elevates Pgc1α protein level and mitochondrial DNA content and enhances mitochondrial activity in mouse and human adipocytes. Taken together, our findings indicate that Parkin regulates mitochondrial homeostasis by balancing mitophagy and Pgc1α-mediated mitochondrial biogenesis in white adipocytes, suggesting a potential therapeutic target in adipocytes to combat obesity and obesity-associated disorders.

Mitochondrial dysfunction contributes to the pathogenesis of metabolic disorders such as obesity[1]. Mitochondria are highly dynamic organelles that play essential roles in energy metabolism, heat production, generation of oxygen radicals, calcium signaling, and apoptosis[2–6]. To maintain metabolic health, mitochondrial selective autophagy (mitophagy), mitochondrial biogenesis, and various mitochondrial protein degradation processes coordinately control the quality of mitochondria[6,7].

Mitophagy involves the recognition of damaged mitochondria by the autophagosome through microtubule-associated protein 1 light chain 3 (LC3) adapters in ubiquitin-dependent and independent mechanisms[8,9]. PTEN-induced putative kinase 1 (Pink1, encoded by *Park6*) and E3 ubiquitin-protein ligase Parkin are known to function in ubiquitin-dependent mitophagy, while other proteins such as TBC1 domain family member 15 (Tbc1d15) and Nip3-like protein X (Nix) contribute to ubiquitin-independent mitophagy[9–12].

Emerging evidence has shown that low mitochondrial content and activity contribute to adipocyte hypertrophy and metabolic disorders[13–18]. Therefore Parkin-mediated mitophagy in adipocytes has attracted increasing attention. Numerous studies indicate that Parkin not only protects mitochondrial function against metabolic stress induced by obesity[19], but also plays an essential role in adipose tissue browning and beige adipocyte thermogenesis[20–22]. However, how adipose tissue Parkin maintains mitochondrial health and regulates glucose homeostasis in adipose tissue remain inadequately understood[20,21,23].

Here we sought to determine whether adipose Parkin regulates cellular metabolism and adiposity. To address this, we generated *Park2*

knockdown 3T3-L1 preadipocytes (Parkin[KD]) and a mouse model by deleting the *Park2* gene specifically in adipose tissue (Parkin[Adi]). We determined that adipose-specific *Park2* deletion prevents both high-fat diet (HFD) and aging-induced obesity. Mechanistic studies revealed that Parkin deletion slightly impaired mitophagy in adipocytes and elevated mitochondrial superoxide levels, which increased the protein level of Nqo1. Moreover, Nqo1 enhances the protein stability of Pgc1α and accelerates mitochondrial biogenesis. Collectively, our findings suggest that Parkin regulates mitochondrial homeostasis in white adipocytes and Parkin is a potential therapeutic target for combating obesity.

## Results

### Parkin is elevated during adipogenesis

Mitophagy, the degradation of mitochondria by autophagic machinery, controls beige adipocyte maintenance[21]. However, the role of mitophagy in white adipocytes and how it relates to adiposity are not well elucidated. We identified over 70 single nucleotide polymorphisms (SNPs) near or within the human *PARK2* gene (encodes Parkin protein) positively associate with BMI in a series of genome-wide association studies (GWAS) from PhenoScanner (a database of human genotype-phenotype associations)[24,25]. Therefore, we examined whether mitophagy-related genes *Park2* and *Park6* are altered during murine adipocyte differentiation. We found that both *Park2* and *Park6* genes were markedly elevated during the differentiation of 3T3-L1 and10T1/2 cells as well as in primary adipocytes of iWAT and brown adipose tissue (BAT) (Fig. 1a, b, Supplementary Fig. 1a, b). Consistently, Parkin protein levels were increased in mature adipose tissues (iWAT and eWAT) and differentiated adipocytes in multiple fat depots (Fig. 1c–e). To determine whether Parkin level is elevated during diet-induced adipogenesis, we fed both male and female wild-type (WT) mice with HFD for 6 weeks. Parkin protein level was only elevated in eWAT of male, but not female WT mice (Fig. 1f; Supplementary Fig. 1c). Increased *Park2* gene expression during the differentiation of 3T3-L1 cells indicates potentially enhanced Parkin-mediated mitophagy. However, this finding was not accordant with elevated mitochondrial DNA (mtDNA) content that we and others have found (Supplementary Fig. 1d)[26,27]. These potentially conflicting findings suggest that Parkin

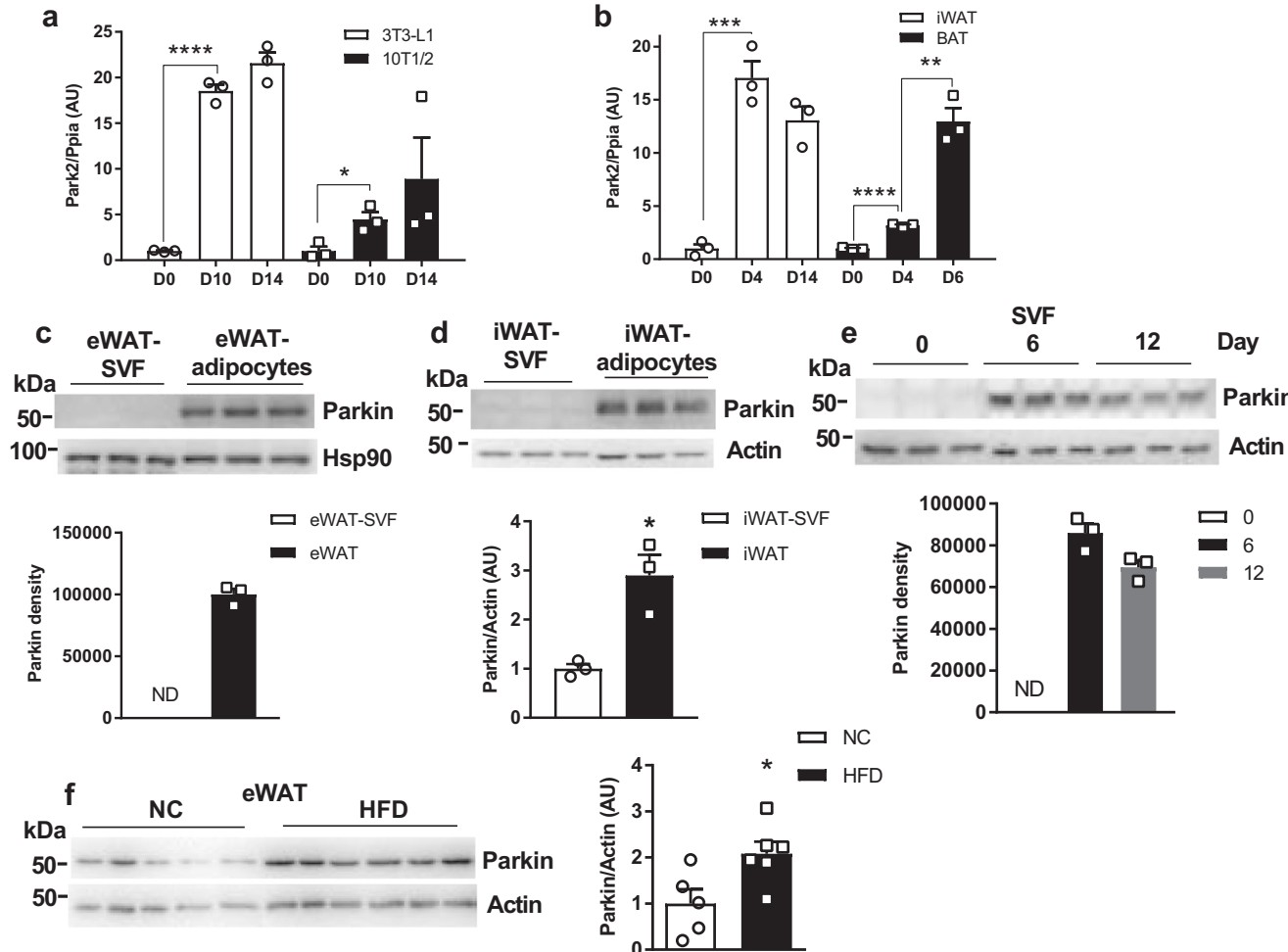

**Fig. 1 | Parkin is elevated in white adipocytes during adipogenesis and by HFD feeding of mice. a**, **b** mRNA level of *Park2* during differentiation of 3T3-L1 and 10T1/2 adipocytes and primary adipocytes from iWAT and BAT of WT mice (*n* = 3 biological replicate) [Unpaired Student's *t* test two-tailed. For 3T3-L1 D0 vs. D10, *p* < 0.0001, 95%CI = 15.58–19.50, R squared = 0.9936; for 10T1/2 D0 vs. D10 *p* = 0.0207, 95%CI = 0.8711–6.061, R squared = 0.7747; for iWAT D0 vs. D4, *p* = 0.0006, 95%CI = 11.56–20.54, R squared = 0.9610, for BAT D0 vs. D4, *p* < 0.0001, 95%CI = 1.913–2.452, R squared = 0.9922; for BAT D4 vs. D6, *p* = 0.0015, 95% CI = 6.284–13.26, R squared = 0.9379.] Western blot analysis (top of panel) and corresponding densitometric quantification (bottom of panel; normalized to Actin) of Parkin in (**c**) eWAT, (**d**) iWAT vs. SVF fractions [Unpaired Student's *t* test two-tailed, for iWAT-SVF vs. iWAT, *p* = 0.0113, 95%CI = 0.7122–3.087, R squared = 0.8314.], (**e**) differentiated primary adipocytes from eWAT SVF fractions at day 0, 6, and 10 (*n* = 3 WT mice per group). **f** Western blot analysis of Parkin in eWAT from normal chow-fed vs. HFD-fed male WT C57BL6/J mice. The bar graph on the right is the densitometric quantification of Parkin protein level normalized to Actin (*n* = 5 mice for NC group, *n* = 6 for HFD group) [Unpaired Student's *t* test two-tailed. For NC vs. HFD, *p* = 0.0254, 95%CI = 0.1670–1.997, R squared = 0.4429.]. Data are presented as mean ± SEM. * *p* < 0.05, ** *p* < 0.01, *** *p* < 0.001, **** *p* < 0.0001. ND = not detectable. AU = arbitrary units. Source data are provided as a Source Data file.

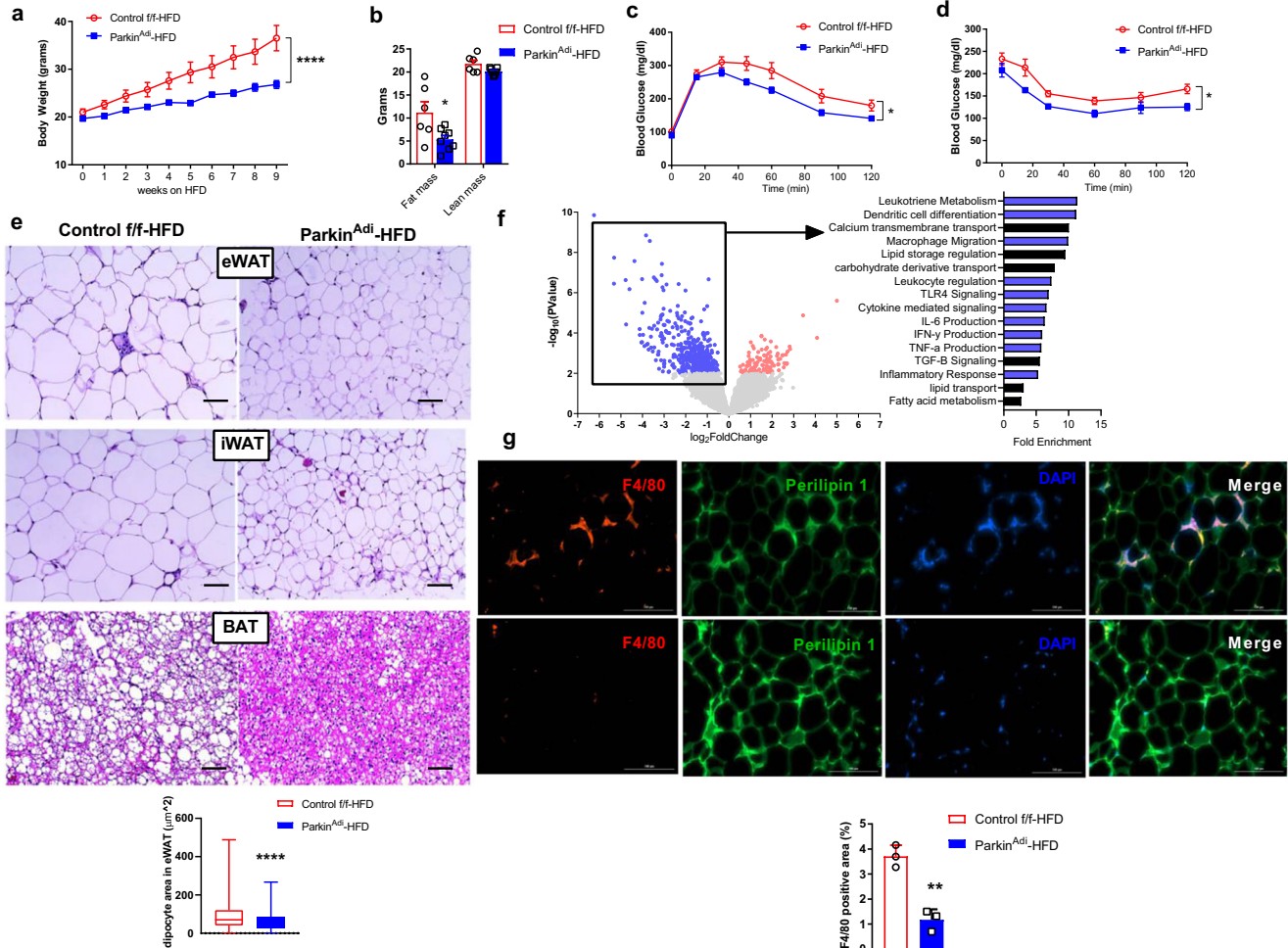

**Fig. 2 | Adipose-specific deletion of Parkin ameliorates high-fat diet and aging-induced obesity and glucose tolerance. a** Body weight of HFD-fed Control f/f and Parkin[Adi] mice ($n = 8$ mice per genotype) [Two-way ANOVA $F_{(1,14)} = 29.51$, $p < 0.0001$, 95%CI = 3.887−6.236]. **b** Body consumption of Control f/f and Parkin[Adi] mice following a 9-week HFD ($n = 6$ mice for Control f/f group, $n = 8$ mice for Parkin[Adi] group) [Unpaired Student's $t$ test two-tailed. For fat mass Control f/f-HFD vs. Parkin[Adi]-HFD, $p = 0.0246$, 95%CI = −10.68 to −0.8768, R squared = 0.3547]. **c** Glucose tolerance test on 6 weeks of HFD-fed Control f/f and Parkin[Adi] mice ($n = 7$ mice per genotype) [Two-way ANOVA $F_{(12)} = 6.477$, $p = 0.0257$, 95% CI = 5.134−66.21]. **d** Insulin tolerance test on 7 weeks of HFD-fed Control f/f and Parkin[Adi] mice ($n = 5$ mice per genotype) [Two-way ANOVA $F_{(1,8)} = 7.594$, $p = 0.0248$, 95%CI = 5.337−60.06]. **e** H&E-stained sections of eWAT, iWAT, and BAT

from 9 weeks HFD-fed mice; scale bars = 50 μm ($n = 5$ mice per genotype). Adipocyte area in eWAT was shown in box and whiskers (Control f/f: minimum value 0.0025, average value 90.50, maximum value 488.2454; Parkin[Adi]: minimum value 0.0025, average value 62.48, maximum value 267.0925) [Unpaired Student's $t$ test two-tailed. $p < 0.0001$, 95%CI = −35.78 to −20.24, R squared = 0.0544]. **f** Volcano plots and enrichment analysis of RNA sequencing results from eWAT of HFD-fed Parkin[Adi] vs. Control f/f mice ($n = 4$ mice per genotype). **g** Immunofluorescent staining of F4/80, Perilipin 1, and DAPI in the eWAT of HFD-fed Control f/f and Parkin[Adi] mice ($n = 3$ mice per genotype) [Unpaired Student's $t$ test two-tailed. $p = 0.0020$, 95%CI = −3.519 to −1.558, R squared = 0.9281]. Data are presented as mean ± SEM. * $p < 0.05$, ** $p < 0.01$, **** $p < 0.0001$. Source data are provided as a Source Data file.

may be involved in the dynamic regulation of mitochondrial homeostasis during adipogenesis.

## Parkin deletion in adipose tissue prevents HFD and aging-induced obesity

To investigate the role of Parkin in adipogenesis, we studied both female and male whole-body Parkin knockout mice (Parkin[KO]). Only male Parkin[KO] mice showed reduced body weight and fat mass (Supplementary Fig. 2a–d), indicating a sexual dimorphism with respect to the role of Parkin in regulating fat mass. Therefore, we will exert the following experiments on male mice. To further explore the role of Parkin in male adipose tissue, we generated adipose tissue-specific *Park2* knockout mice (Parkin[Adi]). Parkin protein levels were dramatically reduced in the eWAT of Parkin[Adi] mice but not in iWAT, BAT, liver, and skeletal muscle, although adiponectin cre was highly expressed in eWAT, iWAT, and BAT (Supplementary Fig. 2e, f). We found that body weight and fat mass were significantly lower than the control f/f mice

after HFD feeding, suggesting that Parkin depletion in adipose tissue is protective against diet-induced obesity (Fig. 2a, b; Supplementary Fig. 2g, h). Both glucose tolerance and insulin tolerance were improved in Parkin[Adi] mice (Fig. 2c, d). Moreover, histological analysis of the adipose tissue indicates a reduction in adipocyte size in eWAT, iWAT, and BAT (Fig. 2e).

To understand the impact of Parkin deletion on the transcriptome in white adipose tissue, we performed RNA-seq and enrichment analysis on eWAT from HFD-fed control f/f and Parkin[Adi] mice. The downregulated genes were enriched in immune-related pathways and biological processes, including macrophage migration and leukocyte regulation (Fig. 2f). Indeed, we confirmed reduced macrophage infiltration in the eWAT of Parkin[Adi] mice by immunofluorescent staining of F4/80 (Fig. 2g). Gene expression of several pro-inflammatory factors including *Tnfα*, *Mcp1*, and *F4/80* in eWAT as well as pro-inflammatory cytokine IL-6 in the plasma were dramatically reduced in Parkin[Adi] vs. control f/f mice (Supplementary Fig. 2i, j).

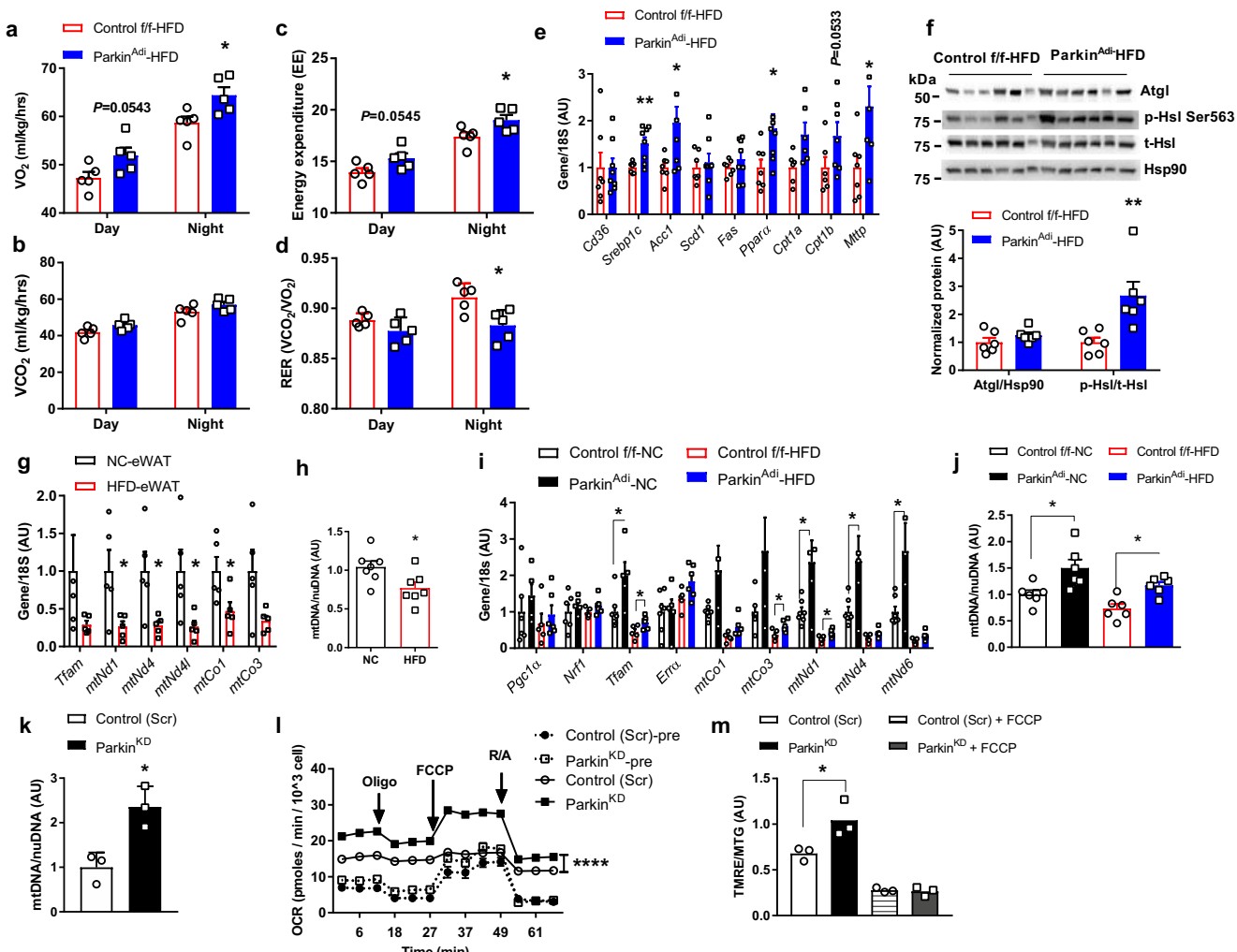

**Fig. 3 | Parkin deletion elevates energy expenditure and mitochondrial activity in adipocytes. a** Oxygen consumption [Unpaired Student's $t$ test two-tailed. For VO2 night, $p = 0.0266$, CI95% $= -0.04921$ to $-0.006597$, R squared $= 0.5327$] (**b**) CO2 production, (**c**) energy expenditure (EE) [Unpaired Student's $t$ test two-tailed. For EE night, $p = 0.0366$, 95%CI $= 0.1279–3.085$, R squared $= 0.4397$] and (**d**) respiratory exchange ratio (RER) [Unpaired Student's $t$ test two-tailed. For RER night, $p = 0.0166$, 95%CI $= -0.04921$ to $-0.006597$, R squared $= 0.5327$] of male Control f/f and Parkin$^{Adi}$ mice fed an HFD for 4 weeks ($n = 5$ mice per genotype). **e** mRNA levels of lipid metabolism genes in eWAT of HFD-fed Control f/f and Parkin$^{Adi}$ mice ($n = 7$ mice for Control f/f group, $n = 8$ mice for Parkin$^{Adi}$ group) [Unpaired Student's $t$ test two-tailed. For Srebp1c, $p = 0.0045$, 95% CI $= 0.1890–0.8345$, R squared $= 0.4744$; for Acc1, $p = 0.0245$, 95%CI $= 0.1441–1.772$, R squared $= 0.3322$; for Ppara, $p = 0.0391$, 95%CI $= 0.04892–1.625$, R squared $= 0.2882$; for Mttp, $p = 0.0216$, 95%CI $= 0.07500–2.348$, R squared $= 0.3101$]. **f** Western blots and densitometric analysis of Atgl, phospho-Hsl Ser563, and total Hsl protein in eWAT of HFD-fed Control f/f and Parkin$^{Adi}$ mice ($n = 6$ mice per genotype) [Unpaired Student's $t$ test two-tailed. For p-HSL/t-HSL, $p = 0.0094$, 95% CI $= 0.5094–2.829$, R squared $= 0.5070$]. **g** mRNA levels of mitochondrial and related genes in the eWAT of WT mice fed with NC and HFD ($n = 5$ mice per diet) [Unpaired Student's $t$ test two-tailed. For mtNd1, $p = 0.0346$, 95%CI $= -1.403$ to $-0.06850$, R squared $= 0.4470$; for mtNd4, $p = 0.0251$, 95%CI $= -1.325$ to $-0.1159$, R squared $= 0.4856$; for mtNd4l, $p = 0.0377$, 95%CI $= -1.417$ to $-0.05340$, R squared $= 0.4360$; for mtCo1, $p = 0.0446$, 95%CI $= -1.045$ to $-0.01642$, R squared $= 0.4145$]. **h** Relative mtDNA copy number in the eWAT of mice fed with NC and HFD ($n = 7$ per diet) [Unpaired Student's $t$ test two-tailed. $p = 0.0274$, 95%CI $= -0.5089$ to $-0.03584$, R squared $= 0.3443$]. **i** mRNA levels of mitochondrial and related genes in eWAT of Control f/f and Parkin$^{Adi}$ mice fed with NC and HFD ($n = 7$ mice for Control f/f-NC group, $n = 5$ mice for Parkin$^{Adi}$-NC group, $n = 5$ mice for Control f/f-HFD

group, $n = 6$ mice for Parkin$^{Adi}$-HFD group) [Unpaired Student's $t$ test two-tailed. For Tfam Control f/f-NC vs. Parkin$^{Adi}$-NC, $p = 0.0218$, 95%CI $= 0.1784–1.819$, R squared $= 0.4240$; for Tfam Control f/f-HFD vs. Parkin$^{Adi}$-HFD, $p = 0.0282$, 95% CI $= 0.03684–0.5148$, R squared $= 0.4310$; for mtCo3 Control f/f-HFD vs. Parkin$^{Adi}$-HFD, $p = 0.0463$, 95%CI $= 0.004537–0.4377$, R squared $= 0.3721$; for mtNd1 Control f/f-NC vs. Parkin$^{Adi}$-NC, $p = 0.0240$, 95%CI $= 0.2213–2.511$, R squared $= 0.4141$, for mtNd1 Control f/f-HFD vs. Parkin$^{Adi}$-HFD, $p = 0.0194$, 95%CI $= 0.03481–0.3077$, R squared $= 0.4725$; for mtNd4 Control f/f-NC vs. Parkin$^{Adi}$-NC, $p = 0.0397$, 95% CI $= 0.08034–2.712$, R squared $= 0.3585$; for mtNd6 Control f/f-NC vs. Parkin$^{Adi}$-NC, $p = 0.0298$, 95%CI $= 0.2004–3.138$, R squared $= 0.3907$]. **j, k** Relative mtDNA copy number in the eWAT of Control f/f and Parkin$^{Adi}$ mice fed with NC and HFD ($n = 6$ mice per genotype per diet) [Unpaired Student's $t$ test two-tailed. For Control f/f-NC vs. Parkin$^{Adi}$-NC, $p = 0.0196$, 95%CI $= 0.09883–0.9015$, R squared $= 0.4354$; for Control f/f-HFD vs. Parkin$^{Adi}$-HFD, $p = 0.0024$, 95%CI $= 0.1960–0.6775$, R squared $= 0.6203$] and in the differentiated 3T3-L1 Control (Scr) and Parkin$^{KD}$ adipocytes ($n = 3$ biological replicate) [Unpaired Student's $t$ test two-tailed. $p = 0.0144$, 95%CI $= 0.4457–2.267$, R squared $= 0.8104$]. **l** Mitochondrial oxygen consumption rate of preadipocytes (dash line) and differentiated adipocytes (solid line) of 3T3-L1 cells ($n = 4$ biological replicate) [Two-way ANOVA F $(1,6) = 198.4$, $p < 0.0001$, 95%CI $= -8.155$ to $-5.741$]. **m** Normalized mitochondrial membrane potential (TMRE)/mitochondrial tracker green (MTG) of the differentiated 3T3-L1 Control (Scr) and Parkin$^{KD}$ adipocytes treated with vehicle (DMSO) and FCCP ($n = 3$ biological replicate) [Unpaired Student's $t$ test two-tailed. For Control (Scr) vs. Parkin$^{KD}$, $p = 0.0424$, 95%CI $= 0.01999–0.7033$, R squared $= 0.6835$]. Data are presented as mean ± SEM. AU = arbitrary units. Oligo = oligomycin, FCCP = trifluoromethoxy carbonylcyanide phenylhydrazone, R/A = rotenone/antimycin a. * $p < 0.05$, ** $p < 0.01$, *** $p < 0.001$. Source data are provided as a Source Data file.

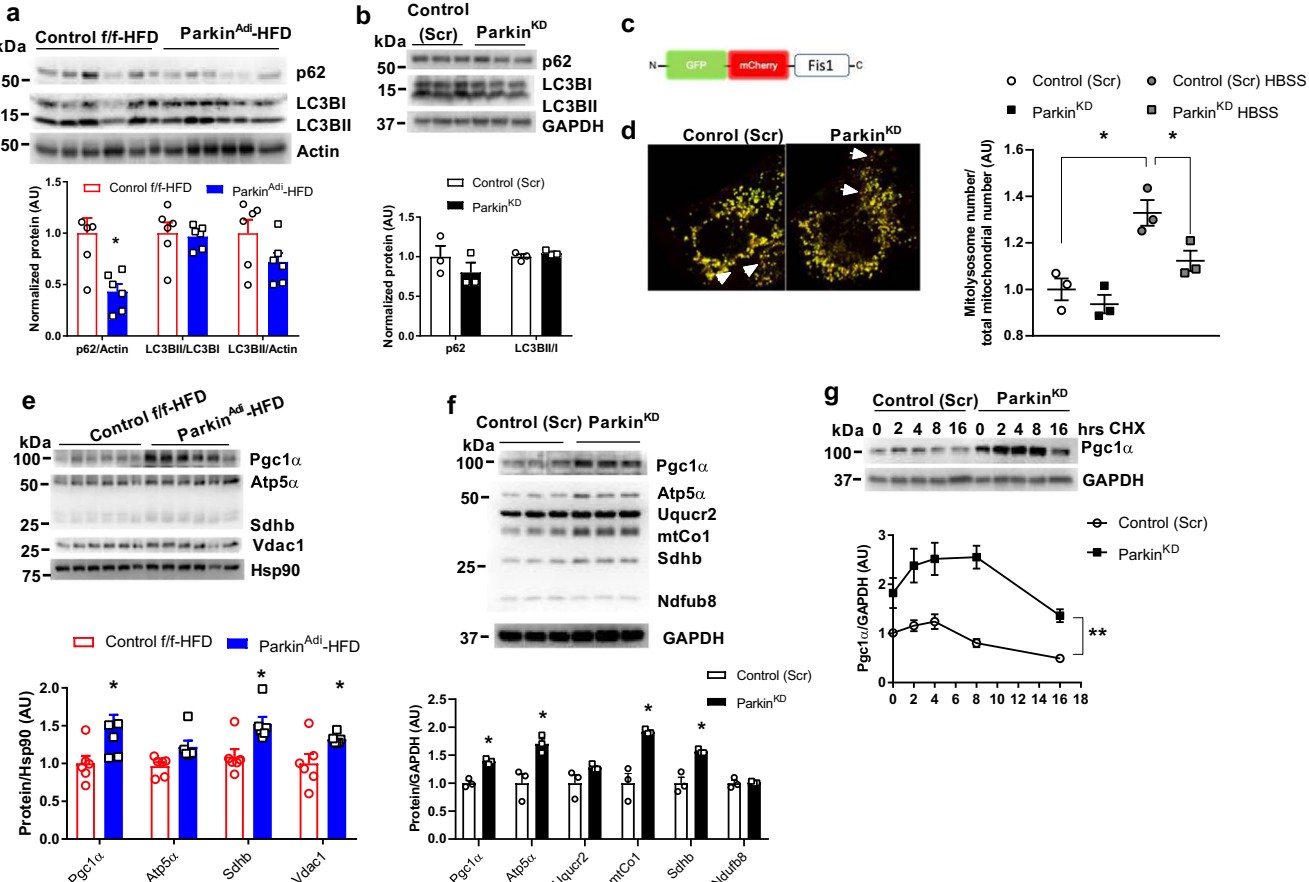

**Fig. 4 | Parkin deletion slightly impairs mitophagy but extended Pgc1α protein stability in adipocytes. a,b** Western blot analysis of autophagy markers p62 and LC3 in the eWAT of Control f/f and Parkin[Adi] mice fed with HFD (*n* = 6 mice per genotype) and in differentiated adipocytes of Control (Scr) and Parkin[KD] (*n* = 3 biological replicate). The bar graphs at the bottom are the densitometric quantification of protein level normalized to Actin or GAPDH [Unpaired Student's *t* test two-tailed. For p62/Actin Control f/f-HFD vs. Parkin[Adi]-HFD, *p* = 0.0065, 95%CI = −0.9381 to −0.1982, R squared = 0.5394]. **c** Adenoviral GFP and mCherry Fis1 probe was used for mitophagy quantification. **d** Fluorescent microscopy analysis of mitophagy signals in Control (Scr) and Parkin[KD] adipocytes. The graph on the right is the quantification of mitolysosome number in differentiated 3T3-L1 Control (Scr) and Parkin[KD] adipocytes cultured in the normal medium or in HBSS for 4 h (*n* = 3 biological replicate) [Unpaired Student's *t* test two-tailed. For Control (Scr) vs. Control (Scr) HBSS, *p* = 0.0106, 95%CI = 0.1274–0.5309, R squared = 0.8369; for Control (Scr) HBSS vs. Parkin[KD] HBSS, *p* = 0.0426, 95%CI = −0.4017 to −0.01107, R squared = 0.6827]. **e** Western blot and densitometry analysis of Pgc1α, OXPHOS

proteins, and Vdac1 (normalized to Hsp90) in eWAT of Control f/f and Parkin[Adi] mice fed with HFD (*n* = 6 mice per genotype) [Unpaired Student's *t* test two-tailed. For Pgc1α, *p* = 0.0446, 95%CI = 0.01360–0.9214, R squared = 0.3450; for Sdhb, *p* = 0.0103, 95%CI = 0.1254–0.7323, R squared = 0.4979; for Vdac1, *p* = 0.0338, 95% CI = 0.02981–0.6094, R squared = 0.3765]. **f** Western blot and densitometry analysis of Pgc1α and OXPHOS proteins in the differentiated Control (Scr) and Parkin[KD] adipocytes (*n* = 3 biological replicate) [Unpaired Student's *t* test two-tailed. For Pgc1α, *p* = 0.0018, 95%CI = 0.2430–0.5338, R squared = 0.9322; for Atp5α, *p* = 0.0201, 95%CI = 0.1810–1.225, R squared = 0.7776; for mtCo1, *p* = 0.0060, 95% CI = 0.4386–1.399, R squared = 0.8759; for Sdhb, *p* = 0.0061, 95% CI = 0.2690–0.8624 R squared = 0.8751]. **g** Western blot and densitometry analysis of Pgc1α (normalized to GAPDH) in the differentiated 3T3-L1 Control (Scr) and Parkin[KD] adipocytes treated with 100 μg/ml CHX at the indicated time. (*n* = 3 biological replicate) [Two-way ANOVA F (1,4) = 21.27, *p* = 0.0099, 95%CI = −1.900 to −0.4720]. Data are presented as mean ± SEM. AU = arbitrary units, CHX = cycloheximide. * *p* < 0.05, ** *p* < 0.01.

Parkin-mediated mitophagy is pivotal for metabolic health[28,29]. Although Parkin[Adi] mice show reduced fat weight gain during HFD-feeding compared to control f/f mice, the long-term effects of Parkin deletion under normal dietary conditions are unknown. To examine the impact of adipose tissue Parkin deletion throughout life, we investigated whether this genetic manipulation would impact aging-associated weight gain. Parkin[Adi] mice showed a reduction in body weight throughout adulthood without perturbation of lifespan (Supplementary Fig. 2k). Although adipocyte size was not altered between genotypes, Parkin[Adi] mice showed improved glucose and insulin tolerance at 9 months old but not 5 months old (Supplementary Fig. 2l–o). Taken together, adipose tissue-specific Parkin deletion prevents both HFD-feeding and aging-induced adiposity and disrupted glucose homeostasis.

## Loss of Parkin in BAT fails to improve glucose homeostasis

Cold stress stimulates mitophagy in brown adipose tissue (BAT)[22,30,31]. Consistently, we found cold stress strikingly increased *Ucp1* gene expression but decreased *Park6* and *Park2* gene expression in BAT (Supplementary Fig. 3a). Because we found Parkin[Adi] mice had higher body temperature at 2 months of age (Supplementary Fig. 3b), we would like to illustrate the role of Parkin in BAT using brown adipose tissue-specific Parkin knockout mouse model (Parkin[BAT]). Parkin protein levels were reduced in BAT but not in other tissues (Supplementary Fig. 3c). Although mtDNA content was significantly increased in BAT of Parkin[BAT] mice, fat mass and glucose homeostasis were not different between genotypes of mice fed with NC or HFD (Supplementary Fig. 3d–g), suggesting that Parkin expression in BAT does not contribute to the lean phenotype and improved glucose homeostasis observed in Parkin[Adi] and Parkin[KO] mice.

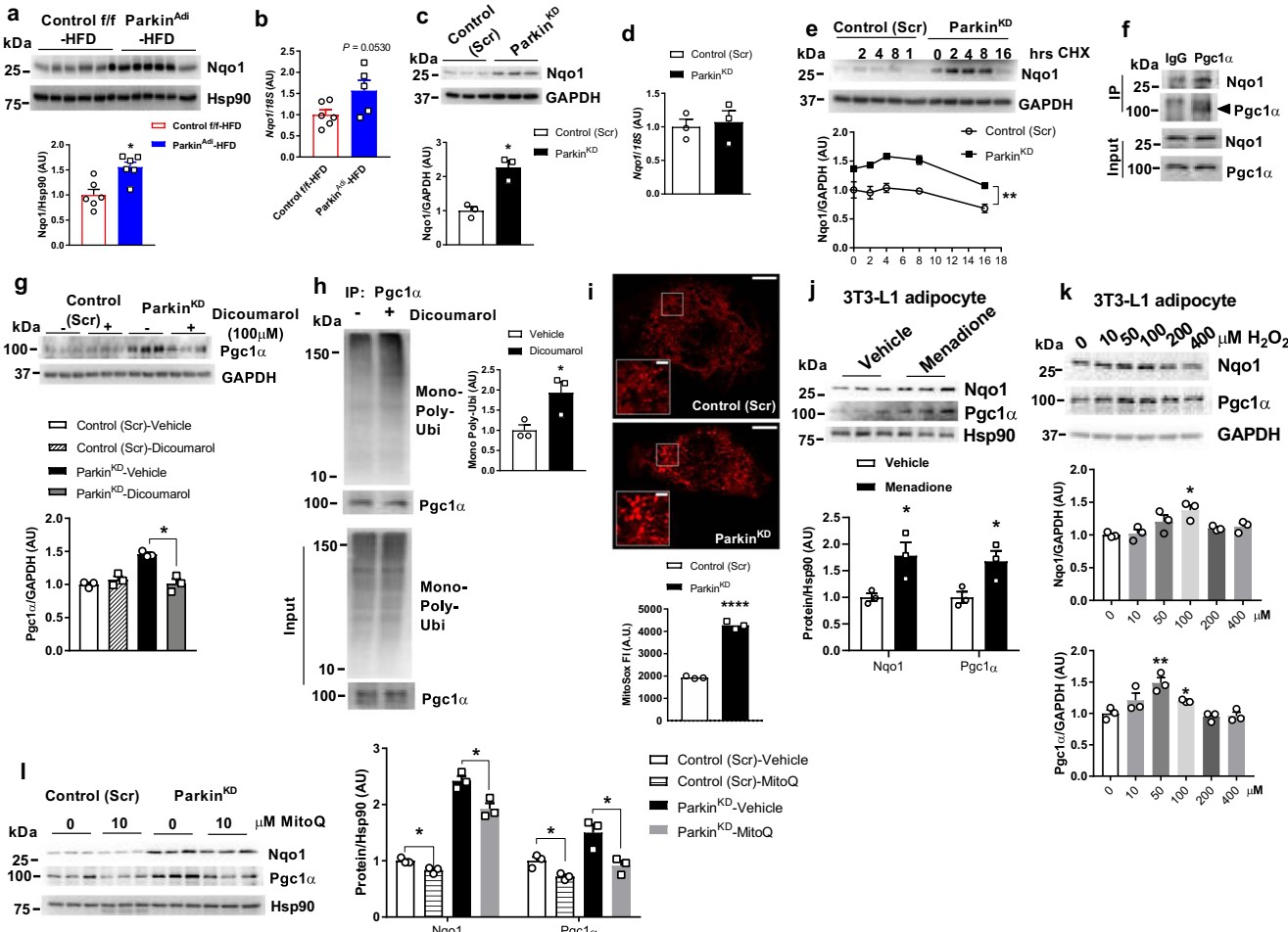

**Fig. 5 | Adipose-specific Parkin deletion enhances Pgc1α protein stability by elevating Nqo1. a,c** Western blot and densitometry analysis of Nqo1 in eWAT of Control f/f and Parkin[Adi] mice fed with HFD (normalized to Hsp90; $n = 6$ mice for each genotype) and in differentiated adipocytes of Control (Scr) and Parkin[KD] ($n = 3$ biological replicate; normalized to GAPDH) [Unpaired Student's $t$ test two-tailed. For Control f/f-HFD vs. Parkin[Adi]-HFD, $p = 0.0040$, 95%CI = 0.2219–0.8880, R squared = 0.5795; for Control (Scr) vs. Parkin[KD], $p = 0.0049$, 95%CI = 0.6432–1.893, R squared = 0.8881]. **b, d** mRNA level of *Nqo1* in eWAT of Control f/f and Parkin[Adi] mice fed with HFD ($n = 6$ mice for each genotype, except one sample in Parkin[Adi]-HFD group with undetectable Nqo1 gene expression) and in differentiated adipocytes of Control (Scr) and Parkin[KD] ($n = 3$ biological replicate). **e** Western blot and densitometry analysis of Nqo1 (normalized to GAPDH) in the differentiated 3T3-L1 Control (Scr) and Parkin[KD] adipocytes treated with 100 μg/ml CHX at the indicated time. ($n = 3$ biological replicate) [Two-way ANOVA F (1,4) = 60.63, $p = 0.0015$, 95% CI = −0.6296 to −0.2986]. **f** Co-immunoprecipitation analysis of Pgc1α and Nqo1 in the lysates of eWAT from WT mice ($n = 1$). **g** Western blot and densitometry analysis of Pgc1α (normalized to GAPDH) in the differentiated 3T3-L1 Control (Scr) and Parkin[KD] adipocytes treated with vehicle and Dicomarol (Nqo1 inhibitor) ($n = 3$ biological replicate) [Unpaired Student's $t$ test two-tailed. For Parkin[KD]-vehicle vs. Parkin[KD]-Dicoumarol, $p = 0.0041$, 95%CI = −0.6501 to −0.2355, R squared = 0.8979]. **h** Co-immunoprecipitation of Pgc1α in differentiated 3T3-L1 Control (Scr) and Parkin[KD] adipocytes treated with vehicle or Dicoumarol ($n = 3$ biological replicate) [Unpaired Student's $t$ test two-tailed. For vehicle vs. Dicoumarol, $p = 0.0379$, 95% CI = 0.08481–1.780, R squared = 0.6999]. **i** Images of MitoSox staining and quantification of MitoSox fluorescent intensity in Control (Scr) and Parkin[KD] adipocytes ($n = 3$ biological replicate, scale bars zoom out = 10 μm, scale bars zoom in = 2 μm) [Unpaired Student's $t$ test two-tailed. For Control (Scr) vs. Parkin[KD], $p < 0.0001$, 95%CI = 2032–2634, R squared = 0.9914]. **j** Western blot and densitometry analysis of Pgc1α and Nqo1 (normalized to Hsp90) in 3T3-L1 adipocytes treated with vehicle and 5 μM Menadione for 24 h ($n = 3$ biological replicate) [Unpaired Student's $t$ test two-tailed. For Nqo1, $p = 0.0392$, 95%CI = 0.06308–1.502, R squared=0.6951; for Pgc1α, $p = 0.0391$, 95%CI = 0.05499–1.295, R squared = 0.6955]. **k** Western blot and densitometry analysis of Pgc1α and Nqo1 (normalized to GAPDH) in 3T3-L1 adipocytes treated with $H_2O_2$ at the indicated concentration for 24 h ($n = 3$ biological replicate) [Unpaired Student's $t$ test two-tailed. For Nqo1 0 vs. 100 μM, $p = 0.0113$, 95%CI = 0.1417–0.6123, R squared = 0.8318; for Pgc1α 0 vs. 50 μM, $p = 0.0089$, 95%CI = 0.2022–0.7678, R squared = 0.8501; for Pgc1α 0 vs. 100 μM, $p = 0.0258$, 95%CI = 0.03867–0.3526, R squared = 0.7496]. **l** Western blot and densitometry analysis of Pgc1α and Nqo1 (normalized to Hsp90) in Control (Scr) and Parkin[KD] adipocytes treated with vehicle or MitoQ (10 μM) for 24 h ($n = 3$ biological replicate) [Unpaired Student's $t$ test two-tailed. For Nqo1 Control (Scr)-vehicle vs. Control (Scr)-MitoQ, $p = 0.0187$, 95%CI = −0.2876 to −0.04575, R squared = 0.7855; for Nqo1 Parkin[KD]-vehicle vs. Parkin[KD]-MitoQ, $p = 0.0188$, 95% CI = −0.8614 to −0.1363, R squared = 0.7849; for Pgc1α Control (Scr)-vehicle vs. Control (Scr)-MitoQ, $p = 0.0207$, 95%CI = −0.4925 to −0.07071, R squared = 0.7745; for Pgc1α Parkin[KD]-vehicle vs. Parkin[KD]-MitoQ, $p = 0.0274$, 95% CI = −1.068 to −0.1072, R squared = 0.7425]. Data are presented as mean ± SEM. AU = arbitrary units. * $p < 0.05$, ** $p < 0.01$. Source data are provided as a Source Data file.

## Parkin deletion increases energy expenditure and mitochondrial activity

To elucidate the mechanisms by which Parkin controls metabolism, we performed indirect calorimetry and found increased oxygen consumption and energy expenditure but reduced RER during the dark phase in body weight-matched Parkin[Adi] vs. control f/f mice (Supplementary Fig. 4a, b). This occurred independent of food intake, water consumption, and physical activity (Fig. 3a–d; Supplementary Fig. 4c–i).

Reduced RER in Parkin[Adi] mice suggests an increased utilization of lipid for energy production. This observation led us to further analyze the lipid metabolism in the adipose tissues. We found that lipid

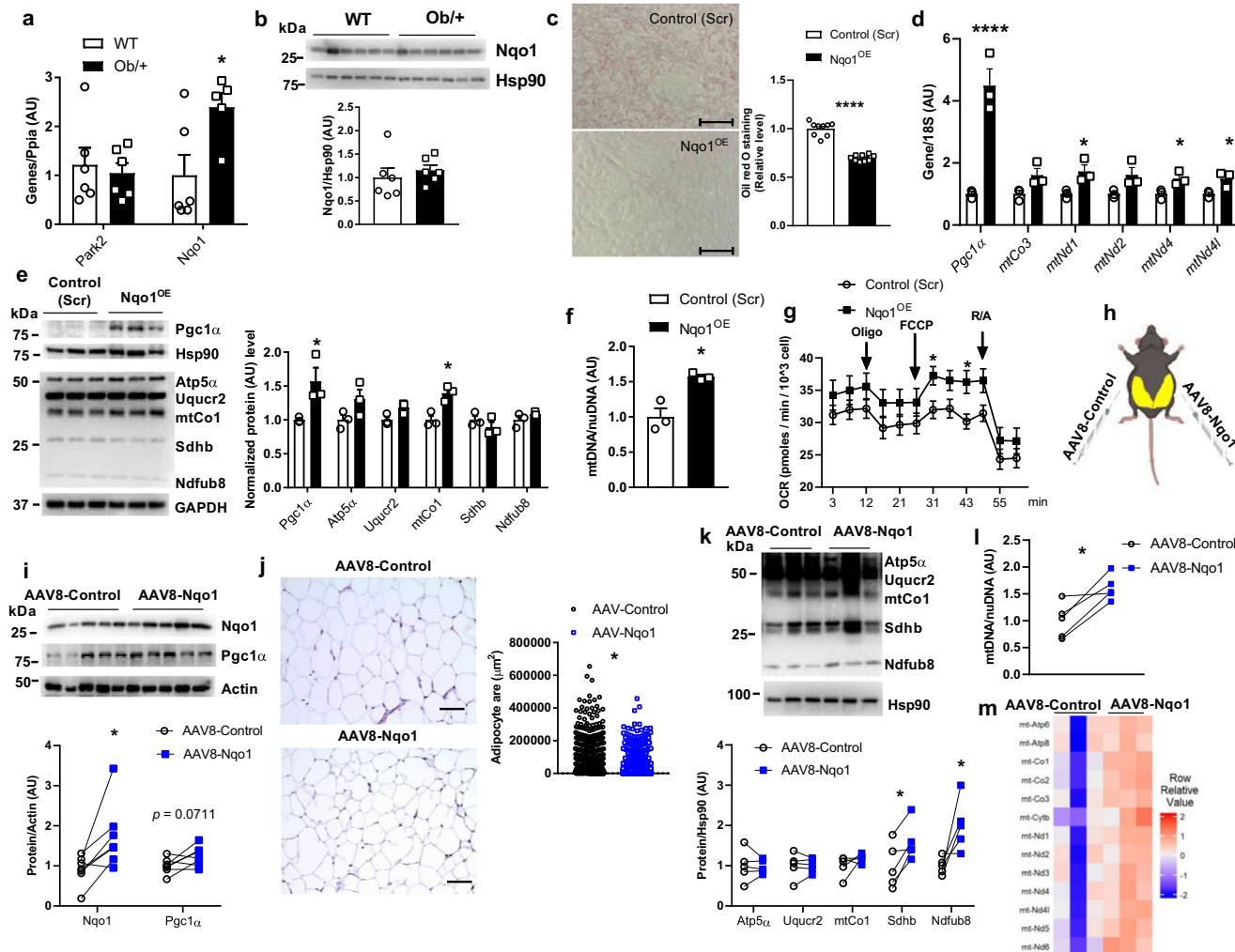

**Fig. 6 | Overexpression of Nqo1 enhances mitochondrial biogenesis in vitro and in vivo. a** mRNA level of *Park2* and *Nqo1* in eWAT of WT and Ob/+ mice (*n* = 6 mice for each group, except one sample in Ob/+ group with undetectable Nqo1 gene expression) [Unpaired Student's *t* test two-tailed. For Nqo1 WT vs. Ob/+, *p* = 0.0276, 95%CI = −0.1921 to 2.596, R squared = 0.4334]. **b** western blot analysis of Nqo1 in eWAT of WT and Ob/+ mice (*n* = 6 mice for each group). **c** Oil Red O staining and the quantification of Oil Red O in differentiated 3T3-L1 Control (Scr) and Nqo1^OE adipocytes (*n* = 9 biological replicate, scale bars = 200 μm) [Unpaired Student's *t* test two-tailed. *p* < 0.0001, 95%CI = −0.3572 to −0.2469, R squared = 0.8939]. **d** mRNA level of mitochondrial and related genes in the differentiated 3T3-L1 Control (Scr) and Nqo1^OE adipocytes (*n* = 3 biological replicate) [Unpaired Student's *t* test two-tailed. For *Pgc1α*, *p* = 0.0030, 95%CI = 1.988–5.001, R squared = 0.9120; for *mtNd1*, *p* = 0.0333, 95%CI = 00.09336–1.359, R squared = 0.7175; for *mtNd4 p* = 0.0341, 95% CI = 0.06033–0.9301, R squared = 0.7142; for *mtNd4l*, *p* = 0.0362, 95% CI = 0.05190–0.9380, R squared = 0.7063]. **e** Western blot and densitometry analysis of Pgc1α and OXPHOS proteins (normalized to Hsp90 or GAPDH) in the differentiated 3T3-L1 Control (Scr) and Nqo1^OE adipocytes (*n* = 3 biological replicate) [Unpaired Student's *t* test two-tailed. For Pgc1α, *p* = 0.0475, 95%CI = 0.01028–1.129, R squared = 0.6665; for mtCo1, *p* = 0.0120, 95%CI = 0.1436–0.6442, R squared = 0.8267]. **f** mtDNA content of the differentiated 3T3-L1 Control (Scr) and Nqo1^OE adipocytes (*n* = 3 biological replicate) [Unpaired Student's *t* test two-tailed. *p* = 0.0108, 95%CI = 0.2180–0.9195, R squared = 0.8351]. **g** Mitochondrial oxygen consumption rate of the differentiated 3T3-L1 Control (Scr) and Nqo1^OE adipocytes (*n* = 5 biological replicate). [Unpaired Student's *t* test two-tailed. For 31 min Control

(Scr) vs. Nqo1^OE, *p* = 0.0278, 95%CI = 00.7418−9.814, R squared = 0.4737; For 43 min Control (Scr) vs. Nqo1^OE, *p* = 0.0200, 95%CI = 1.247−10.99, R squared = 0.5119]. **h** Experimental design for AAV8 injection in HFD-fed male WT mice. Created with BioRender.com. **i** Western blot and densitometry analysis of Nqo1 and Pgc1α proteins (normalized to Actin) in eWAT of AAV8-Control or AAV8-Nqo1 injected mice (*n* = 7 mice per group) [Paired Student's *t* test two-tailed, *t* = 3.058. For Nqo1, *p* = 0.0223, 95%CI = 0.1673–1.508, R squared = 0.6091]. **j** H&E-stained sections of the eWAT from AAV8-Control or AAV8-Nqo1 injected mice; scale bars = 50 μm; The dot graph on the right is the quantified adipocyte area from H&E-stained sections of the eWAT from HFD-fed Control f/f and Parkin^Adi mice (*n* = 6 mice for AAV8-Control group, *n* = 7 for AAV8-Nqo1 group, total1603-1932 adipocytes) [Unpaired Student's *t* test two-tailed. *p* = 0.0296, 95%CI = −9156 to −476.6, R squared = 0.001345]. **k** Western blot and densitometry analysis of OXPHOS protein levels (normalized to Hsp90) in eWAT of AAV-Control or AAV8-Nqo1 injected mice (*n* = 5 mice per group) [Paired Student's *t* test two-tailed. For Sdhb, *t* = 3.916, *p* = 0.0173, 95% CI = 0.1716–1.008, R squared = 0.7931; for Ndufb8, *t* = 2.886, *p* = 0.0447, 95% CI = 0.03842–1.988, R squared = 0.6755]. **l** mtDNA content of the eWAT from AAV8-Control or AAV8-Nqo1 injected mice (*n* = 5 mice per group) [Paired Student's *t* test two-tailed. *t* = 4.023, *p* = 0.0158, 95%CI = 0.1896–1.034, R squared = 0.8018]. **m** Heatmap of mtDNA encoded genes from RNA sequencing results in AAV8-Control or AAV8-Nqo1 injected eWAT (*n* = 3 mice per group). Data are presented as mean ± SEM. AU = arbitrary units. Oligo = oligomycin, FCCP = trifluoromethoxy carbonylcyanide phenylhydrazone, R/A = rotenone/antimycin a. * *p* < 0.05, *** *p* < 0.001. Source data are provided as a Source Data file.

transfer gene *Mttp*, lipid synthesis genes (*Srebp1c* and *Acc1*), and fatty acid oxidation genes (*Pparα*, *Cpt1α*, *Cpt1b*) were significantly increased in eWAT of Parkin^Adi vs. control f/f mice (Fig. 3e). In addition, the protein levels of phosphorylated hormone-sensitive lipase (HSL) at

Ser563 were robustly elevated in the eWAT of Parkin^Adi mice (Fig. 3f). Collectively, Parkin deletion alters the gene expression that impact lipid synthesis and oxidation in eWAT and could suggest alterations in mitochondrial oxidative metabolism.

We and others have shown that mitochondrial mass and function are impaired in the context of obesity and type-2 diabetes in humans and rodents[1,15,32] (Fig. 3g, h; Supplementary Fig. 4j). Furthermore, we found that both mtDNA encoded genes and mtDNA copy number were elevated regardless of diet or fat pad depot in Parkin[Adi] compared with control f/f mice (Fig. 3i, j; Supplementary Fig. 4k, l). To explore the impact of Parkin deletion on mitochondrial function, we knocked down the *Park2* in 3T3-L1 cells (Parkin[KD]) and found mtDNA copy number was increased 2.3 folds in Parkin[KD] vs. control (Scr) adipocytes (Supplementary Fig. 4o; Fig. 3k). Increased mtDNA copy number was also observed in the eWAT of whole-body Parkin knockout mice (Parkin[KO]) (Supplementary Fig. 4m, n, 4p).

Consistent with our mouse studies, Parkin[KD] adipocytes also displayed a reduction in lipid levels (Supplementary Fig. 4q). Moreover, increased mitochondrial oxygen consumption rate in both Parkin[KD] preadipocyte and adipocytes measured by the Seahorse XF analyzer indicates mitochondrial function was enhanced as a consequence of Parkin reduction (Fig. 3l). We also observed a dramatic increase in mitochondrial membrane potential and aspect ratio (Fig. 3m; Supplementary Fig. 4r), reflecting a healthy mitochondrial population in Parkin[KD] adipocytes. Taken together, our in vitro and in vivo findings confirm our observation in mice that reduction in Parkin expression promotes mitochondrial function in adipocytes.

### Parkin mediates both mitophagy and mitochondrial biogenesis in adipocytes

Mitochondrial content is regulated by a balance between mitophagy and mitochondrial biogenesis[8]. Because mitophagy relies on the autophagic machinery and our data suggests that lacking Parkin impacts mitochondrial abundance and function, we next examined the protein levels of autophagy markers p62 and LC3BI/II. Only p62 protein was reduced but no change in the protein levels of LC3BI and LC3BII in eWAT of HFD-fed Parkin[Adi] vs. control f/f mice (Fig. 4a, b). Although mitophagy heavily relies on macro-autophagy, alterations in macro-autophagy makers do not necessarily mean that mitophagy is impacted. Therefore, we decided to directly monitor and quantify the mitophagy using GFP-mCherry-Fis1 (mito-QC reporter). Red fluorescent protein (RFP) signal is typically maintained despite the acidic lysosomal environment (pH 4–5) whereas green fluorescent protein (GFP) loses fluorescence in this pH range[33,34]. Using the adenoviral GFP-mCherry-Fis1, we assessed the turnover of mitochondria by lysosome and observed a decreased number of mitolysosome puncta indicating a reduction of mitophagy in Parkin[KD] vs. control (Scr) adipocytes (Fig. 4c, d). Next, we assessed the protein levels of mitochondrial biogenesis markers. Surprisingly, the protein level of Pgc1α was strikingly elevated in both eWAT of HFD-fed Parkin[Adi] mice and Parkin[KD] adipocytes compared with controls (Fig. 4e, f). The protein levels of voltage-dependent anion selective channel 1 (Vdac1) and subunit b of succinate dehydrogenase (Sdhb) were also significantly elevated, but Parkin-interacting substrate (Paris) was not altered in Parkin[Adi] mice (Fig. 4e; Supplementary Fig. 5a). Because the gene expression of Pgc1α was not changed in eWAT of both NC and HFD-fed Parkin[Adi] mice (Fig. 3i), we hypothesized that Parkin deletion may promote post-translational modification of Pgc1α to enhance the protein stability. Next, we blocked protein synthesis with cycloheximide (CHX) and determined that the protein level of Pgc1α was elevated in Parkin[KD] adipocytes (Fig. 4g). These findings suggest that Parkin may control mitochondrial homeostasis by coordinating mitophagy with mitochondrial biogenesis in adipocytes.

### Elevated Nqo1 stabilizes Pgc1α protein in Parkin knockdown adipocytes

To identify the mechanisms underlying Pgc1α protein stability in the context of Parkin knockdown, we investigated the impact of Nqo1 expression on Pgc1α in adipocytes, as Nqo1 was previously shown to modulate Pgc1α protein[35]. Both the protein and mRNA levels of Nqo1 were elevated in eWAT of HFD-fed Parkin[Adi] mice and Parkin[KD] adipocytes (Fig. 5a–d). Similar to Pgc1α, Nqo1 protein stability was also significantly elevated in Parkin[KD] adipocytes following treatment with the protein synthesis inhibitor cycloheximide (CHX) (Fig. 5e).

Nqo1 is known to interact with Pgc1α in an NADH-dependent manner[35]. To confirm the protein association of Pgc1α and Nqo1 in adipose tissues, we performed co-immunoprecipitation using eWAT protein lysates of WT mice. We determined that Pgc1α associates with Nqo1 in white adipose tissues (Fig. 6f). To further confirm Nqo1 preserves the protein level of Pgc1α, we next tested whether pharmacological inhibition of Nqo1 activity or genetical knocking down Nqo1 in adipocytes could decrease the protein level of Pgc1α. We treated adipocytes with the Nqo1 inhibitors Dicoumarol or ES936 and observed that Pgc1α protein was decreased in Parkin[KD] adipocytes compared with untreated (Fig. 5g; Supplementary Fig. 5b). Nqo1 and Parkin double knock down reduced the protein level of Pgc1α in Parkin[KD] preadipocytes, although this difference did not reach statistical significance (Supplementary Fig. 5c). Furthermore, we found Nqo1 inhibitor Dicoumarol increased the protein ubiquitination of Pgc1α in 3T3-L1 adipocytes (Fig. 5h; Supplementary Fig. 5d). Together, our findings suggest Pgc1α associates with Nqo1 in white adipose tissue and increased Nqo1 protein stabilizes Pgc1α protein in adipocytes with reduced Parkin expression.

### Mitochondrial superoxide increases Nqo1 and Pgc1α in Parkin deleted adipocytes

Because Nqo1 is induced by oxidative stress[36–38], we examined oxidative stress in Parkin[KD] adipocytes and determined that superoxide levels were significantly increased (Fig. 5i; Supplementary Fig. 5e). Since mitochondria are a primary source of superoxide, we treated Parkin[KD] adipocytes with Menadione, a redox-cycling quinone that produces toxic levels of mitochondrial superoxide. We found that menadione increased the protein levels of both Nqo1 and Pgc1α in adipocytes (Fig. 5j). Moreover, $H_2O_2$ administration could also elevate the protein levels of both Nqo1 and Pgc1α in 3T3-L1 adipocytes at 50–100 μM (Fig. 5k). However, N-acetyl cysteine (NAC), which generates hydropersulfides with direct antioxidation activity, did not change the protein levels of Nqo1 and Pgc1α in adipocytes (Supplementary Fig. 5f). MitoQ, a mitochondrial-matrix-targeted antioxidant, suppressed the protein levels of both Nqo1 and Pgc1α (Fig. 5l; Supplementary Fig. 5g). In response to low-dose of oxidative stress, Nrf2 induces the expression of cytoprotective genes such as Nqo1[39]. Indeed, Nrf2 gene expression is elevated in eWAT of Parkin[Adi] mice (Supplementary Fig. 5h), suggesting elevated superoxide activates Nrf2-induced Nqo1, which enhances Pgc1α protein stability in Parkin deleted adipocytes.

### Nqo1 overexpression elevates Pgc1α protein level in adipocytes

We and others have found that Nqo1 gene expression is positively associated with adiposity in humans and mice (Supplementary Figs. 6a and 7a,b)[40]. However, a recent study shows that whole-body Nqo1 transgenic mice prevented HFD-induced obesity and insulin resistance[38], suggesting Nqo1 gene expression may not be concordant with its protein level. We then examined both gene expression and protein level of Nqo1 in Ob/+ mice, a genetically obese mouse model. Indeed, we found that Nqo1 gene expression, but not the protein level, was significantly increased in eWAT of Ob/+ mice (Fig. 6a, b). Therefore, we carefully draw any conclusions from gene association results. To investigate the impact of Nqo1 expression on mitochondrial function and lipid metabolism, we overexpressed Nqo1 in 3T3-L1 adipocytes (Nqo1[OE]). We observed a reduction of lipid levels that corresponded with an increased expression of mtDNA-encoded genes, Pgc1α mRNA and protein levels, mitochondrial proteins, mtDNA content, and mitochondrial oxygen consumption rate in Nqo1[OE]

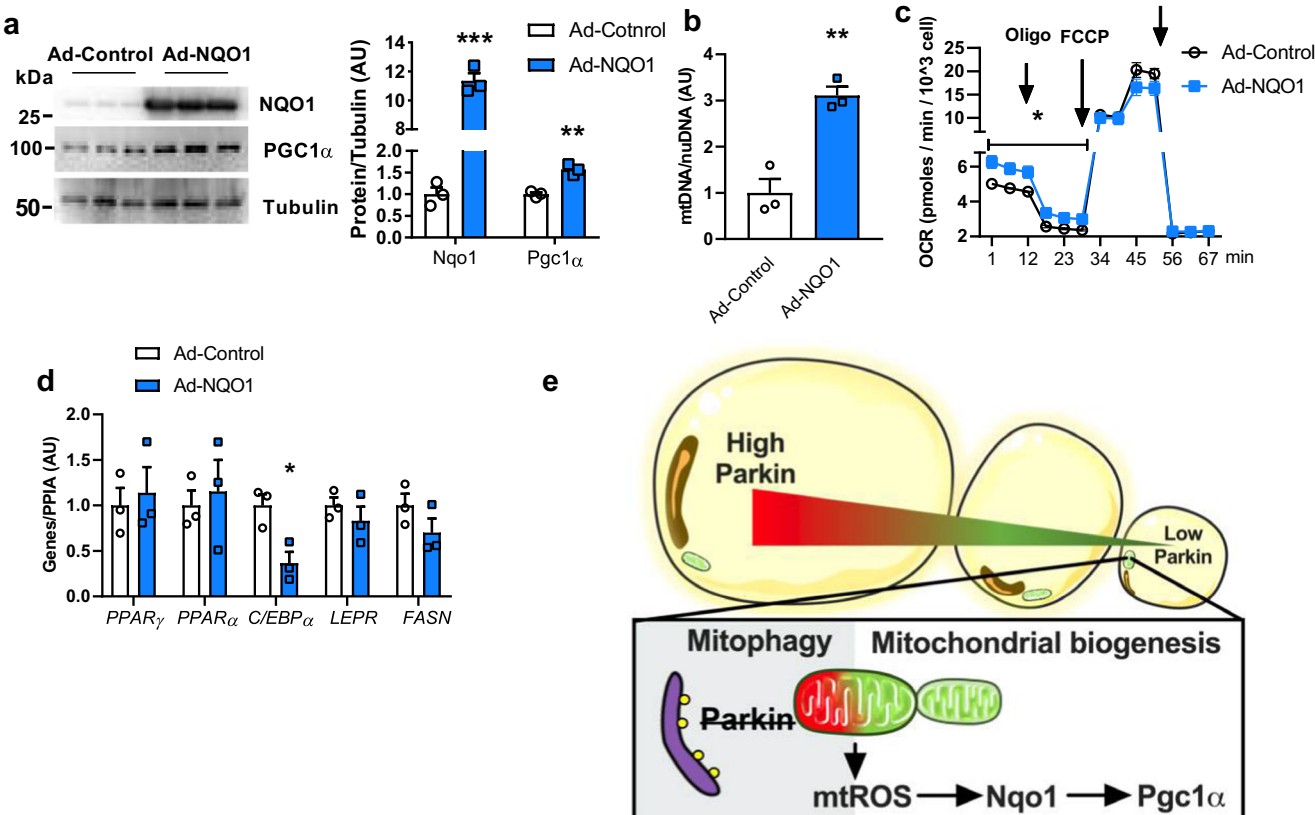

**Fig. 7 | Overexpression of NQO1 in human SubQ adipocytes elevates mtDNA content and improves mitochondrial function. a** Western blot and densitometry analysis of Nqo1 and Pgc1α protein levels in ad-Control and ad-NQO1 treated primary human subcutaneous adipocytes ($n = 3$ biological replicate) [Unpaired Student's $t$ test two-tailed. For Nqo1, $p < 0.0001$, 95%CI = 18.824–11.90, R squared = 0.9887; for Pgc1α, $p = 0.0024$, 95%CI = 0.3372–0.7968, R squared = 0.9215]. **b**–**d** mtDNA content, mitochondrial oxygen consumption rate, and adipogenesis gene expression of ad-Control and ad-NQO1 treated primary human subcutaneous adipocytes ($n = 3$ biological replicate) [Unpaired Student's $t$ test two-tailed. For mtDNA Ad-Control vs. Ad-NQO1, $p = 0.0041$, 95%CI = 1.119–3.099, R squared = 0.8973; for OCR 1–29 min, Ad-Control vs. Ad-NQO1, two-way ANOVA F (1,4) = 12.83, $p = 0.0231$, 95%CI = −1.630 to −0.2065].or Pgc1α, $p = 0.0024$, 95% CI = 0.3372–0.7968, R squared = 0.9215, for C/EBPα Ad-Control vs. Ad-NQO1, $p = 0.0221$, 95%CI = −1.118 to −0.1497, R squared = 0.7677]. **e** Schematic view of Parkin regulates adiposity by coordinating mitophagy with mitochondrial biogenesis in white adipocytes. Created with BioRender.com. Data are presented as mean ± SEM. AU = arbitrary units. Oligo = oligomycin, FCCP = trifluoromethoxy carbonylcyanide phenylhydrazone, R/A = rotenone/antimycin a. * $p < 0.05$, ** $p < 0.01$, *** $p < 0.001$. Source data are provided as a Source Data file.

adipocytes (Fig. 6c–g; Supplementary Fig. 6b). Nqo1 overexpression strikingly altered mitochondrial morphology by diminishing mitochondrial aspect ratio (the ratio between centerline length and average width) (Supplementary Fig. 6c). To determine the impact of Nqo1 overexpression upon adipogenesis, we profiled adipogenesis gene expression and found a dramatic suppression of *C/Ebpα*, *Fabp4*, *Adiponectin*, and *Leptin* in Nqo1[OE] vs. control (Scr) adipocytes (Supplementary Fig. 6d). In addition, both Nqo1 protein stability and mitophagy were elevated, while protein synthesis was decreased in Nqo1[OE] adipocytes (Supplementary Fig. 6e–g).

To explore the role of Nqo1 in vivo, we contralaterally injected AAV8-Control or AAV8-Nqo1 into eWAT of HFD-fed WT mice so that each mouse served as its own control (Fig. 6h). AAV8-Nqo1 significantly elevated Sdhb and Ndufb8 protein levels, mtDNA content, mtDNA encoded genes, and decreased adipose tissue size (Fig. 6l, m, Supplementary Fig. 6h, i).

### Overexpression of NQO1 in human adipocytes increases PGC1α protein and improves mitochondrial respiration

To determine the role of NQO1 in human adipocytes, we overexpressed human NQO1 in differentiated human subcutaneous adipocytes with adenoviral NQO1-IRES-GFP (Ad-NQO1). Ad-NQO1 not only robustly elevated NQO1 protein but also increased the protein level of

PGC1α (Fig. 7a; Supplementary Fig. 7c). Consistent with the findings from Nqo1[OE] adipocyte and AAV8-Nqo1 injected mice, NQO1 overexpression in human adipocytes also significantly elevated mtDNA content and basal mitochondrial activity and mildly decreased lipid levels (Fig. 7b, c; Supplementary Fig. 7d, e). Furthermore, elevated NQO1 also suppressed the adipogenesis gene *C/Ebpα* (Fig. 7d). Altogether, our findings suggest that NQO1 controls mitochondrial function and *C/Ebpα* expression that contribute to the inhibition of adipogenesis in human adipocytes.

## Discussion

In this study, we provide evidence that the E3 ubiquitin-protein ligase Parkin controls mitochondrial homeostasis in human and rodent white adipocytes. Beyond its regulatory role in mitophagy, Parkin prevents the turnover of the mitochondrial biogenesis master regulator Pgc1α through Nqo1 in white adipocytes. Stabilized Pgc1α protein drives increased mitochondrial abundance and function which combats HFD- and aging-induced obesity.

Mitochondrial homeostasis, regulated by a balance of mitophagy and mitochondrial biogenesis, involves in the pathogenesis of obesity and other metabolic disorders[8]. Consistent with other published work, we obtained the paradoxical result that both mitochondrial number and mitophagy protein Parkin were increased during adipogenesis

in vitro and in vivo[41]. Herein, we investigated this apparent inconsistent observation by studying Parkin-regulated mitochondrial homeostasis and whole-body metabolism using an adipose tissue-specific *Park2* deletion mouse model. We show that Parkin deletion ameliorates HFD-feeding and aging-induced adiposity and improves glucose homeostasis. Supporting the lean phenotype that we observed in the Parkin[Adi] mouse model, we consistently found smaller adipocytes, suppressed inflammation, increased energy expenditure, and elevated mitochondrial activity from multiple cohorts.

Parkin-mediated mitophagy has been shown to play an important role in the beige-to-white adipocyte transition[20,21,42]. Our findings agree with published work showing that cold stress downregulates *Park2* gene level in BAT[22]. Multilocularity of brown adipocytes and higher body temperature of Parkin[Adi] mice led us to hypothesize that Parkin deletion in BAT may contribute to the lean phenotype and improved glucose homeostasis. We tested this notion by deleting *Park2* gene specifically in BAT. Unfortunately, Parkin[BAT] mice did not recapitulate the phenotypes we observed in Parkin[Adi] mice, where Parkin was deleted from both white and brown adipocytes. Thus, our findings suggest that Parkin expression in BAT may be dispensable in the regulation of adiposity.

Pgc1α regulates mitochondrial biogenesis by activating a variety of transcription factors[43–45], which could benefit adipose tissue and systemic metabolism[46,47]. Our in vitro and in vivo studies suggest that Parkin deletion elevates Pgc1α protein stability. The increase in Pgc1α led to a direct increase in markers of mitochondrial biogenesis including elevated mtDNA copy number and the level of mitochondrial gene expression[48]. Consistent with previous pre-clinical and clinical findings, we found increased mtDNA content and mtDNA encoded gene expression in Parkin-deleted adipocytes, suggesting that loss of Parkin specifically in adipose tissue prevented diet-induced obesity[49]. Our findings highlight that Parkin, in addition to controlling mitophagy, plays a central role in regulating mitochondrial biogenesis in the prevention of adipocyte hypertrophy. Mitochondrial biogenesis reversely regulates mitophagy signal, where Pgc1α and NRF1 transcriptionally regulate the mitophagy receptor FUNDC1 to enhance mitophagy[50]. Together, these findings indicate the complex interplay between mitochondrial biogenesis and mitophagy in maintaining mitochondrial homeostasis[30].

Numerous studies have shown that pharmacological stimulation of Nqo1 or whole-body Nqo1 overexpression ameliorates obesity[38,51–55]. We confirmed and extended these findings by showing that Nqo1 overexpression selectively in white adipocytes enhances mitochondrial activity and lipid metabolism. Our findings provide an opportunity for further studies to examine the metabolic regulation of Nqo1 in vivo.

Parkin has been shown to ubiquitinate Paris, which acts as a transcriptional repressor of Pgc1α in dopamine neurons[56]. Herein, Parkin deletion did not change the protein level of Paris in white adipose tissues, indicating that Parkin may exert different mechanisms to regulate Pgc1α in different tissues. A previous study suggested that Nqo1 interacts with Pgc1α in an NADH-dependent manner by inhibiting degradation of Pgc1α, but not canonical ubiquitin-mediated pathways[35]. In our study, we found that Dicoumarol administration increased Pgc1α ubiquitination in adipocytes, suggesting Nqo1 may inhibit Pgc1α degradation through the ubiquitin-dependent pathway. However, we did not rule out that the Dicoumarol treatment may be working through alternative mechanisms to inhibit Pgc1α degradation. A further thorough examination of the mechanisms of Nqo1-mediated Pgc1α degradation is needed.

We showed that Parkin deletion from adipocytes enhanced mitochondrial function by elevating both mitochondrial respiration and mitochondrial membrane potential. A consequence of increased mitochondrial metabolism in Parkin-deleted adipocytes was an elevation of mitochondrial reactive oxygen species (mtROS). The cellular effects of ROS are largely dependent on the species itself, its concentration, and likely time of exposure. ROS are important physiological molecules to regulate intracellular signaling pathways[57]. Studies have shown that NADPH oxidase (NOX) derived ROS are important modulators of adipogenesis and adipose tissue function[58,59]. Our findings suggest that a low concentration of mitochondrial superoxide in Parkin-deleted adipocytes may activate cytoprotective signals to maintain adipocyte health (Fig. 7e).

In conclusion, our research shows that Parkin not only regulates mitophagy but also mitochondrial biogenesis in white adipocytes through inhibiting Nqo1 increased protein stabilization of Pgc1α. These findings suggest that Parkin regulates mitochondrial hemostasis and energy metabolism in WAT, suggesting that Parkin is a viable therapeutic target to combat obesity and obesity-associated disorders.

# Methods

## Animals

All mouse experimentation and animal care were approved by the University of California, Los Angeles Institutional Animal Care and Use Committee (IACUC). Mice were maintained on a 12 h light/dark cycle from 6 am to 6 pm at ambient temperature (~72 F) with controlled humidity (~45%) in pathogen-free conditions. Food consumption, mouse activity and health were monitored daily by the Division of Laboratory Animal Medicine at UCLA. Prior to organ harvest, mice were euthanized by isoflurane overdose followed by cervical dislocation. This is an approved method according to the recommendations of the panel on Euthanasia of the American Veterinary Medical Association. Parkin floxed mice (a gift from Ted Dawson) were crossed with adiponectin Cre mice (The Jackson Laboratory, #010803) or Ucp1 Cre mice (The Jackson Laboratory, #024670) to generate animals with Parkin deletion in adipose tissue specifically (Parkin[Adi]) or in brown adipose tissue separately (Parkin[BAT]). Whole-body parkin null mice were obtained from the Jackson Laboratory (The Jackson Laboratory, #006582). Mice were studied under normal chow (NC, LabDiet 5053) and 45 % high-fat diet (HFD, Research Diets, D12451)-fed conditions. All experiments were performed on the male mouse except the results in Supplementary Fig. 1c and Supplementary Fig. 2c,d were collected from female mice. All these mice were in C57BL/6 J background about 4–6 months old. Blood was drawn from 6 h-fasted male mice and analyzed for circulating factors: glucose (HemoCue), inflammatory cytokines and chemokines (Meso Scale Discovery, CA, Cat # K15048D). Intraperitoneal glucose tolerance tests (IPGTT, 1 g/kg dextrose) were performed on mice after fasting (6 h for normal chow-fed mice, overnight for HFD-fed mice). After one week of recovery intraperitoneal insulin tolerance tests (IPITT, 0.7 U/kg for NC-fed mice, 0.75 U/kg for HFD-fed mice) were performed on these mice with 6 h fasting[32].

## METSIM studies

Genetic association and gene expression analyses were conducted on data collected from the metabolic syndrome in men (METSIM) study as previously published[60]. Gene-trait relationships presented here were obtained from 770 male participants (age of 45–70 years). No new human samples were acquired for the generation of this manuscript.

## Metabolic analysis

Body fat and lean mass were determined by NMR (Bruker). Oxygen consumption, carbon dioxide production, and respiratory exchange ratio were determined in male mice fed with HFD for 4 weeks using the Promethion metabolic screening system (Sable Systems).

## Cell culture and transfections

The stromal vascular fraction (SVF) cells from inguinal white adipose tissue (iWAT), eWAT, or BAT of male C57BL6/J wild-type mice (The Jackson Laboratory, #000664, 5 weeks old) were cultured in DMEM/F12 medium with 10 % FBS as described[61]. Then the cells were cultured

2 days in DM1 medium (DMEM medium, 10% FBS, 5 µg/ml insulin, 1 µM Dexamethasone, 0.5 mM IBMX, and 1 µM rosiglitazone), 2 days in DM2 medium (DMEM/F12 medium with 10% FBS plus 5 µg/ml insulin), and 6 additional days in DMS medium. The primary adipocytes from BAT were differentiated in (DMEM/F12 medium with 10% FBS, 5 µg/ml insulin, 0.5 µM Dexamethasone, 0.125 mM Indomethacin, 1 nM T3, 0.5 mM IBMX, and 1 µM GW1929). 3T3-L1 preadipocytes were cultured cells in DM1 after 100% confluent for 2 days and in DM2 (DMEM medium, 10 % FBS, and 5 µg/ml insulin) for 2 days before replacement with DMS (DMEM medium and 10 % FBS) for 2–3 days. Primary human subcutaneous pre-adipocytes were purchased from ATCC (PCS-210-010) and differentiated to adipocytes according to the protocol provided (ATCC, PCS-500-050). Briefly, pre-adipocytes were incubated for 48 h before initiating differentiation via adipocyte differentiation initiation medium for 96 h. After this period, cells were cultured in adipocyte differentiation maintenance medium for an extra 3 days and then incubated with or without $2 \times 10^7$ PFU/ml adenovirus carrying human NQO1 (Vector Biolabs, ADV-217023) for another 5 days. To achieve Park2 knockdown (KD), lentiviral particles carrying shRNA targeted to Park2 or scramble shRNA (Sigma-Aldrich) (multiplicity of infection = 3) were used to transduce 3T3-L1 pre-adipocytes. To inhibit Nqo1 activity, differentiated adipocytes were treated with the Nqo1 inhibitor (100 µM Dicoumarol, Santa Cruz Biotechnology, CAS66-76-2,) or (400 nM ES936, Santa Cruz Biotechnology, CAS192829-78-3) for 24 h. To overexpress Nqo1 in adipocytes (Nqo1$^{OE}$), 3T3-L1 pre-adipocytes were transfected with pcDNA3-Nqo1-flag (Addgene, #61729) and selected with 500 µg/ml G418 for 4 weeks. Cells were harvested and Nqo1 overexpression was confirmed by immunoblot.

## AAV injection

Male C57BL/6 J wild-type mice (2 months old) fed with HFD for 4 weeks (The Jackson Laboratory, #000664) were anesthetized with isoflurane and fixed into a supine position. The midventral line region was shaved and cleaned with ethanol. A 1.5–2 cm midline incision was made in the skin followed by separation of the underlying musculature to expose the epididymal white adipose tissue (eWAT). $2.5 \times 10^{11}$ GC/µl of AAV8-hAdp-m-Nqo1 (AAV8-Nqo1) (Vector Biolabs, AAV-266028) dissolved in 50 uL of saline was injected directly into the left eWAT. The same amount of control virus (Vector Biolabs, AAV-7077) and saline volume was injected into the other side of eWAT. The body wall was sutured and the wound was closed with wound clips. Animals recovered on a heating pad. To prevent infection, sulfamethoxazole (0.48 mg/ml) and trimethoprim (0.096 mg/ml) via an oral suspension was supplied. Animals were observed daily for any signs of inflammation. 7 days after surgery, the wound clips were removed. Tissues were harvested 4 weeks after virus injection.

## Fluorescence microscopy

All imaging was performed using a Zeiss LSM880 confocal microscope. Super-resolution imaging was performed with a 63× apochromat oil-immersion lens and an AiryScan super-resolution detector. Image analysis was performed using Fiji (ImageJ, NIH)[62]. To quantify mitophagy, adipocytes were transduced with adenoviral mCherry-GFP-Fis1 (a generous gift from Orian Shirihai) for 3 days. The fluorescent signals were captured and analyzed. In studies assessing membrane potential, cells were labeled with MitoTracker Green (MTG; Invitrogen) and Tetramethylrhodamine, Ethyl Ester, Perchlorate (TMRE; Invitrogen). Mitochondrial superoxide was stained by MitoSOX in the live 3T3-L1 adipocytes. Fluorescence was quantified by confocal microscopy and analyzed in Fiji.

## ROS detection

Hydrogen peroxide and superoxide were detected using ROS/superoxide detection assay kit (Abcam, ab139476). Briefly, 3T3-L1 adipocytes were cultured in 96 well plate and incubated in ROS/Superoxide detection solution with or without ROS inducer for 1 h in the dark. The fluorescent signals were detected by fluorescent microplate reader and standard fluorescein (excitation/emission = 488/520 nm) and rhodamine (excitation/emission = 550/610 nm). The signals were normalized to protein content in each well.

## Mitochondrial DNA

Total DNA was extracted from cells using DNeasy Blood and Tissue kit (Qiagen). For mouse adipocytes, primers for mitochondrial encoded cytochrome c oxidase III (mtCO3) for mtDNA and 18 S for nuclear DNA were used to assess mtDNA content. The ratio of mtCO3 to 18 S was used to determine mtDNA content[32]. For human adipocytes, primers for mitochondrially encoded tRNA leucine 1 for mtDNA and beta-2-microglobin (B2M) for nuclear DNA were used to determine mtDNA content. The ratio of mtCO3 to 18 S was used to determine mtDNA content[49].

## Mitochondrial respiration

Mitochondrial respiration in 3T3-L1 preadipocytes and differentiated adipocytes was measured using an XF96 Extracellular Flux Analyzer (Seahorse Biosciences). Briefly, cells were plated and grown to confluence on a 96-well plate. Measurements of oxygen consumption were made continuously while cells were sequentially treated with oligomycin (ATP synthase inhibitor), FCCP (an uncoupling agent), and rotenone/myxothiazol (inhibitors of complex I /III of the electron transport chain).

## Immunoblot analysis

Mouse tissue samples were pulverized in liquid nitrogen and homogenized in RIPA lysis buffer containing freshly added complete EDTA-Free protease (Roche) and Phosphatase Inhibitor Cocktail 2 (Sigma-Aldrich). All lysates were centrifuged and resolved by SDS-PAGE. Samples were transferred to PVDF membranes and subsequently probed with the following antibodies for protein detection: Parkin (Cell Signaling #2132), Pink1 (Cayman Chemicals, #10006283), p62 (Progen Biotechnik GmbH, #03-GP62-C), actin (Santa Cruz Biotechnology, sc-47778), Lc3b (Novus, NB100), GAPDH (Invitrogen, AM4300), F4/80 (Santa Cruz Biotechnology, sc-377009), Pgc1α (EMD Millipore, AB3242; Abcam, ab191838; and Abcam, ab106814), Hsp90 (Cell Signaling #4877), OXPHOS (Abcam, ab110413), Nqo1 (Novus, NB200-209; Abcam, ab80588), p53 (Santa Cruz Biotechnology, sc-126), Mono- and polyubiquitinylated conjugates recombinant monoclonal antibody (ENZO lifeSCIENCES, ENZ-ABS840, Clone UBCJ2), Paris (Abcam, ab130867), Hmgb1 (Abcam, ab18256), α-Tubulin (Santa Cruz Biotechnology, sc-5286), Perilipin 1 (Cell Signaling, #9349), AlexaFluor 488 goat anti rabbit IgG (Invitrogen, A11088), or AlexaFluor 568 goat anti mouse IgG (Invitrogen, A11004). After transfer, PVDF membranes were cut in half and imaged separately by the BIO-RAD ChemiDoc XRS imaging system. The exposure time was adjusted by using the transform settings in the BioRad Quantity One image software (BIO-RAD). The cropped images were organized to form a figure in GraphPad Prism9.4.1 software. Densitometric analysis was performed using BioRad Quantity One image software.

## Co-immunoprecipitation

3T3-L1 adipocytes (~1.7 × 10^7 cells/each) were lysed in the lysis buffer (10 mM Hepes-KOH, pH 7.9, 0.5% NP-40, 140 mM NaCl, 10 mM KCl, 1.5 mM MgCl2 and protease inhibitors) by dounce on ice. The lysate was centrifuged at 16,000 g for 15 min at 4 °C, and the resulting supernatant was used for immunoprecipitation. After preincubation of each supernatant with 20 µl of Protein A/G magnetic beads (Thermo Scientific, Cat #: 88802) at 4 °C for 1 h the pre-cleared supernatant was incubated with 4 µg of normal IgG or anti-Pgc1α antibody and 30 µl Protein A/G magnetic beads at 4 °C overnight

with rotation. After washing 4 times with lysis buffer, the bound proteins were eluted from the beads by boiling them in SDS sample buffer. The eluted proteins were analyzed by Western blotting as described above.

## Protein synthesis assay

Protein synthesis was determined by using Protein Synthesis Assay Kit (Cayman chemical, Cat # 601100). Briefly, differentiated adipocytes were treated with O-Propargyl-Puromycin (OPP) working solution and followed with a cell-based assay fixative buffer. Washed adipocytes were incubated with 5 FAM-Azide staining solution for 30 min at room temperature. Nuclei were stained with Hoechst 33342 for 10 min. FITC (excitation/emission = 485/535 nm) signal was detected by a fluorescent plate reader. Nuclei were counted by the Operetta high-content imaging system (PerkinElmer).

## Immunohistochemistry

Formalin-fixed adipose tissues were sectioned and stained for H&E. Following deparaffinization, rehydration, heat-induced epitope retrieval, and permeabilization, slides were blocked with 5% BSA and incubated with the Perilipin 1 antibody overnight followed by the goat anti-rabbit IgG-Alexa Fluor 488 overnight inside a humidified chamber at 4 °C. After BSA blocking, the slides were incubated with F4/80 antibody overnight and followed by the goat anti-mouse IgG-Alexa Fluor 568 overnight at 4 °C again. Images were captured under a fluorescent microscope (Zesis) and BioTek Lionheart LX automated microscope (Agilent).

## Quantitative RT-PCR

Mouse tissue samples were homogenized using Trizol reagent and RNA was isolated using RNeasy Mini QIAcube Kit with DNAse digestion (Qiagen). For RNA isolation from cells, lyses occurred using RLT buffer (Qiagen) with 2-Mercaptoethanol. RNA was isolated from lysed cells using the RNeasy Plus kit per the manufacturer's instructions. cDNA synthesis was performed using 1 μg of RNA with iSuperscript reverse transcriptase (Bio-Rad). PCR reactions were prepared using PowerUp SYBR Green Master Mix (Applied Biosystems). All PCRs are performed in a QuantStudio 5 Real-Time PCR system (Applied Biosystems). Quantification of a given gene, expressed as relative mRNA level compared with control, was calculated after normalization to a standard housekeeping gene (18 S or ACTIN). Primer pairs were designed using Primer 3 Input software (Version 0.4.0) or previously published sequences. Primer sets were selected spanning at least one exon-exon junction when possible and were checked for specificity using BLAST (Basic Local Alignment Search Tool, NCBI). The specificity of the PCR amplification was confirmed by melting curve analysis ensuring that a single product with its characteristic melting temperature was obtained. See Table S1 for a list of the primers used.

## RNA sequencing and analysis

A homogenous portion of frozen adipose samples were homogenized using a tissue homogenizer in Trizol. RNA was isolated using the Qiagen RNeasy Mini QIAcube Kit following the manufacturer's instructions. Isolated RNA was checked for concentration using a NanoDrop and purity using the Agilent TapeStation. Only samples with RIN > 7.0 were used. Libraries were prepared using the KAPA mRNA HyperPrep Kit following the manufacturer's instructions. The resulting libraries were combined into two pools and sequenced on an Illumina HiSeq 3000 within the UCLA TCGB core facility following in-house established protocols. Raw reads were checked for quality using FastQC, aligned to the Mus musculus GRCm38, and then counted using Mus musculus GRCm38 version 97. Alignment and counting occurred using Rsubread v 2.4.2[63]. Raw counts were then analyzed for differential gene expression using the DESeq2 v1.30.0[64].

## Statistics

Values presented are expressed as means ± SEM unless otherwise indicated. Statistical analyses were performed using Student's $t$ test when comparing two groups of samples or Two-way analysis of variance (ANOVA) for identification of significance within and between groups using GraphPad Prism9.4.1 (GraphPad Software). Significance was set a priori at $P < 0.05$. For RNA Sequencing analysis, the significance was set as FDR < 0.05.

## Reporting summary

Further information on research design is available in the Nature Portfolio Reporting Summary linked to this article.

## Data availability

Source data are provided as a Source Data file. METSIM adipose array data are available from in the Gene Expression Omnibus database under accession code GSE70353, and additional clinical data were obtained from previously published papers[60,65]. The mouse RNA-seq results from eWAT of HFD-fed Parkin[Adi] vs. Control f/f mice and AAV8-Nqo1 vs. AAV8-Control injected eWAT have been deposited in the Gene Expression Omnibus database under accession code GSE207496. Human *PARK2* gene (encodes Parkin protein) association is available from PhenoScanner v2 (http://www.phenoscanner.medschl.cam.ac.uk). The authors declare that all data supporting the findings of this study are available within the paper and its Supplementary Information files.

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

## Acknowledgements

We would like to thank Dr. Ted M. Dawson at Johns Hopkins University for providing us Parkin flox mice. We thank undergraduate students Julia Zhou at the University of California San Diego for analyzing the adipocyte size. We thank Wenjuan Ren for immunoblotting and Calvin Pan for HMDP data analysis. Z.Z. was supported by an NIH grant (DK125354). A.L.H. was supported by NIH grants (DK109724, P30DK063491). A.J.L. was supported by NIH grants (U54 DK120342 and DK117850). T.M.M. was supported by the UCLA Intercampus Medical Genetics Training Program (T32GM008243), NRSA predoctoral fellowship (F31DK108657), Carl V. Gisolfi Memorial Research grant from the American College of Sports Medicine, and a predoctoral graduate student award from the Dornsife College at the University of Southern California. D.M.W. is supported by an EMBO long-term fellowship ALTF 828-2021.

## Author contributions

Z.Z., T.M.M., and A.L.H conceived and designed the experiments. Z.Z., T.M.M., A.R.S., and X.Z. performed all in vivo studies. Z.Z., T.M.M., X.Z., A.R.S., Y.C., A.M., and P.S. conducted the sample collection and subsequent experimental analysis. B.L.C., and T.Q.d.A.V. performed indirect calorimetry studies. Z.Z. and T.M.M. performed all adipocyte culture studies and mtDNA content analysis. L.C. performed all AAV injection studies. D.M.W., J.N., M.S. conducted studies related to mitochondrial respirometry and confocal microscopy. Z.Z., T.M.M., L.C., X.Z., A.R.S., Y.C., B.L.C., A.M., P.S. T.Q.d.A.V., M.L., O.S.S., A.L.H, and A.J.L. analyzed the data. T.M.M., L.C., A.L.H, A.J.L., and Z.Z. wrote the paper.

## Competing interests

The authors declare no competing interests.

## Additional information

[1]Division of Cardiology, Department of Medicine, University of California, Los Angeles, CA 90095, USA. [2]Division of Dermatology, Department of Medicine, University of California, Los Angeles, CA 90095, USA. [3]Division of Endocrinology, Diabetes, and Hypertension, Department of Medicine, University of California, Los Angeles, CA 90095, USA. [4]Division of Pediatric Endocrinology, Department of Pediatrics UCLA Children's Discovery and Innovation Institute, University of California, Los Angeles, CA 90095, USA. [5]Department of Biological Chemistry, David Geffen School of Medicine, University of California, Los Angeles, CA 90095, USA. [6]Molecular Biology Institute, University of California, Los Angeles, CA 90095, USA. [7]Institute of Clinical Medicine, Internal Medicine, University of Eastern Finland, 70210 Kuopio, Finland. [8]Department of Human Genetics, University of California, Los Angeles, CA 90095, USA. [9]Veterans Administration Greater Los Angeles Healthcare System, Geriatric Research Education and Clinical Center (GRECC), Los Angeles, CA, USA. [10]Present address: MRC Mitochondrial Biology Unit, University of Cambridge, Cambridge CB2 0XY, UK. [11]Present address: Department of Clinical Neurosciences, School of Clinical Medicine, University of Cambridge, Cambridge Biomedical Campus, Cambridge, UK. [12]Present address: Department of Endocrinology and Metabolism, Zhongshan Hospital, Fudan University, Shanghai, P. R. China. [13]These authors contributed equally: Timothy M. Moore, Lijing Cheng. ✉e-mail: zhenqizhou@mednet.ucla.edu

