## [Peer Review File · Nature Communications]

Title: Parkin Regulates Adiposity by Coordinating Mitophagy with Mitochondrial Biogenesis in White AdipocytesREVIEWER COMMENTS

Reviewer #1 (Remarks to the Author):

In this manuscript, Moore and colleagues studied the function of Parkin, an E3 ubiquitin-protein ligase, in adipocytes. They found that adipose tissue-specific deletion of the Parkin gene (Park2) can protect mice against HFD induced obesity. The author further suggested that Park2 deletion can elevate mitochondrial biogenesis by increasing PGC1alpha protein stability, which is mediated by Nqo1 protein.

Overall, the studies are very interesting. These conclusions will potentially provide important and novel mechanisms in adipose tissue dysfunction and the development of metabolic disorders. However, there are a few shortcomings in the quality of data and experimental approach that prevent firm conclusions at this point.

Major concerns:

1. Many of the western-blot images are low in quality. For example, the actin bands are weak in Figure 1C, and Parkin showed two different sizes in SVF and adipocytes.
2. All histology images are small, and many of them are very dark, which makes it very hard to tell the morphology.
3. For the immunofluorescent staining in Figure 2G, please co-stain F4/80 with Perilipin 1 to validate live adipocytes.
4. Although the authors made firm conclusions about adipocyte cell size, quantification of cell size was not provided.
5. Electron microscopy images of mitochondria are highly recommended to show the results of altered mitochondrial biogenesis.
6. The conclusion that “Loss of Parkin in BAT fail(s) to prevent HFD-feeding induced obesity” is overstated. There was no description of the method used for brown adipose tissue-specific Parkin knockout. Which mouse model was used? In Figure S3C, the protein levels of Parkin were not only reduced in BAT, but also in eWAT and iWAT, which indicates that it’s not a brown adipocyte-specific model.
7. In the ParkinBAT mice, is there any difference in body weight after HFD feeding? Which time point was used for the glucose tolerance test in Figure S3G?
8. Does Parkin KO also promote mitochondrial biogenesis in iWAT and BAT? If not, why is the mechanism specific to eWAT?
9. The body weight changes for HFD-fed control and ParkinAdi in Figure S4A were not consistent with those in Figure 2A.

Minor concerns:

1. Please provide the age of mice in Figure S2Q.
2. For the IP experiment in Figure 5F, there are bands in the IgG group Figure 5F.

Reviewer #2 (Remarks to the Author):

Moore et al. took advantage of mice on high-fat diet with genetic intervention of Parkin in adipose tissue to elucidate the mechanism by which targeting Parkin in adipocytes prevents obesity by regulating mitochondrial homeostasis through mitophagy and mitochondrial biogenesis. This is clearly one of the most important area with great potential impacts on obesity and metabolic syndromes. The concept is also novel. However, this reviewer has identified many problems with the data ranging from the experimental design, validity of the findings and the quality of the data. Overall, the findings do not warrant the conclusion that Parkin regulates mitochondrial biogenesis and mitophagy through the regulation of Pgc-1 α protein stability via Nqo1.

1. Data presented in Fig. 1 showed that Park2 (Parkin) mRNA increases during adipogenic differentiation in both immortalized cells lines and primary adipocytes as well as in eWAT in mice. These findings establish the potential correlation of Parkin expression with adipose tissue expansion, which is the foundation of this paper. The identity of the band shown as Parkin needs validation. If the bands in eWAT and iWAT (c and d) are true Parkin bands, the expression in SVF should be interpreted as not detectable rather as an arbitrary unit of 1.
2. Fig. 1E is confusing. 0 day was described as SVF fractions from eWAT and 10 day as 10 day-differentiated adipocytes, why it is labeled as eWAT-SVF. The quantification of this panel is not trustworthy.
3. Fig. 2 lacks control. Control and Parkinadi on normal chow diet must be included. GTT and ITT are relatively crude assays for whole body metabolism. This reviewer has not previously seen such tight data from mice with a sample size of 6-8 as presented in Fig. 2C and 2D. Appropriate controls (groups on NC), the GTT and ITT data should be included, which will be revealing. Fig. 2E should be quantified for each mouse. Fig. 2G is difficult to evaluate. Could not see the image. No quantification.
4. Same issue for NC groups for Fig. 3. Fig. 3I showed data from mice on NC. Why not include all data in Fig. 1 and Fig. 2?
5. The data in Fig. 3I suggest that gene deletion of Parkin in adipocytes leads to enhanced gene expression related to mitochondrial biogenesis under NC condition. Fig. 3A-3E showed increased VO₂, VCO₂, and EE with slightly decreased RER in Parkin AKO mice. It is critical to show the locomotive activity during the dark and light cycle. If Parkin AKO mice on HFD have significantly higher cage activities, it could potentially explain all the findings presented in this paper.
6. Most importantly, Parkin AKO needs to be validated. No data show how effective the deletion of the Park2 gene in adipocytes or WAT/BAT in Parkin AKO mice. The same is true for ParkinKD adipocytes.
7. The statistical analysis for Fig. 3M is problematic. Each individual cell cannot be counted as a sample. The mean value of the measured cells in the same well is considered as n = 1. The same issue is also raised for other figures in this manuscript.
8. The data presented in Fig. 4 has the same issue of lacking NC control and the lack of evidence of adipose-specific KO of Parkin.
9. The interpretation of reduced p62 as evidence of reduced autophagy is not warranted. The level of p62 protein is an indication of the level of one of the cargo proteins for autophagy, but itself is degraded

during the process of autophagy. With no other supporting evidence, it is premature to interpret the data as evidence of decreased autophagy.

10. Fig. 4C-4D has the same problem of using the number of cells from the same well as the number of samples. The use of the construct of GFP-mCherry-Fis1 for quantification of mitophagy needs to be validated. The data as presented is not as convincing to suggested impaired mitophagy by Parkin KD.

11. The image of Fig. 4F is cropped. The same size (n=3) is small. Cropping image data needs to be explained.

12. The identity of Pgc-1a protein needs to be validated as it is the focus of the study. Pgc-1a is a very labile protein with a half-life in the range of 1-2 h (Rasbach et al. Arch Biochem Biophys PMID 18718443, 2008; Park et al. PNAS, 2015 PMID 25918410). CHX treatment blocks translation, which should always result in reduced protein abundance over time. It is strange that under both control and KD conditions, Pgc-1a protein increased at least in the first 4 hours.

13. It is also peculiar that CHX treatment causes increased Nqo1 for 8 hours in Fig. 5E. The data presented in Fig. 5H needs to be quantified with a sufficient sample size.

14. The data presented in Fig. 6A does not make logical sense. HFD caused an increased Parkin expression (Fig. 1F), which impaired mitochondrial gene expression (Fig. 3G) and repressed Pgc-1alpha gene expression (Fig. 3I). ParkinAdi mice had increased Nqo1 (Fig. 5A), suggesting that Parkin inhibit Nqo1 expression and/or promote Nqo1 degradation. If anything, Ob/+ mice (more like HFD mice) should have reduced Nqo1 expression, which is the opposite at least for Fig. 6A. A critical piece of data is to show that in the absence of Nqo1 protein induction, Pgc-1alpha protein induction under the condition of Parkin KO/KD is gone to establish the causal relationship although it does not negative the value of the findings for enhanced Nqo1 expression as a therapeutic potential of treating obesity.

We thank all reviewers for the efforts taken to review our manuscript. The comments provided by the reviewers have significantly enhanced the readability, understanding, and strength of our manuscript. We have responded to each point in red in this document and highlighted the corresponding changes in the manuscript.

Here are several changes to our manuscript,

- (1) We removed Figures 1G and S1D. Because we are not able to explain why Parkin gene expression and protein level are not consistent *in vivo*. It is possible regulatory feedback loops are present that we have yet to uncover.
- (2) We removed Suppl. Figure 6C, because mitochondrial membrane potential is not our primary focus in this manuscript.
- (3) We removed Figure 7A to simplify our findings.
- (4) We have fixed an error and updated Suppl. Figure 6A.

Reviewer #1 (Remarks to the Author):

In this manuscript, Moore and colleagues studied the function of Parkin, an E3 ubiquitin-protein ligase, in adipocytes. They found that adipose tissue-specific deletion of the Parkin gene (Park2) can protect mice against HFD induced obesity. The author further suggested that Park2 deletion can elevate mitochondrial biogenesis by increasing PGC1 α protein stability, which is mediated by Nqo1 protein.

Overall, the studies are very interesting. These conclusions will potentially provide important and novel mechanisms in adipose tissue dysfunction and the development of metabolic disorders. However, there are a few shortcomings in the quality of data and experimental approach that prevent firm conclusions at this point.

Major concerns:

1. Many of the western-blot images are low in quality. For example, the actin bands are weak in Figure 1C, and Parkin showed two different sizes in SVF and adipocytes.

We thank the reviewer for pointing out this problem. We repeated the immunoblots and included the new blots in Figures 1C, 1D, and 1E.

2. All histology images are small, and many of them are very dark, which makes it very hard to tell the morphology.

We have increased the brightness of all the histology pictures.

3. For the immunofluorescent staining in Figure 2G, please co-stain F4/8o with Perilipin 1 to validate live adipocytes.

We have included the new co-staining in Figure 2G.

4. Although the authors made firm conclusions about adipocyte cell size, quantification of cell size was not provided.

The quantification of adipocyte size of eWAT was in Suppl. Figure 2. We have moved it under Figure 2E in the revised manuscript.

5. Electron microscopy images of mitochondria are highly recommended to show the results of altered mitochondrial biogenesis.

We agree that electron microscopy images would present high-resolution mitochondria images. However, it only provides a snapshot of mitochondria in a tiny area of the tissue. It is well known that Pgc1a is a hallmark of mitochondrial biogenesis (Cell, 1998, 92(6):829-39. PMID: 9529258;

PNAS, 2012, 109(24):9635-40. PMID: 22645355). Therefore, we performed biochemistry assays to determine the Pgc1a level in the entire whole eWAT lysates.

6. The conclusion that “Loss of Parkin in BAT fail(s) to prevent HFD-feeding induced obesity” is overstated. There was no description of the method used for brown adipose tissue-specific Parkin knockout. Which mouse model was used? In Figure S3C, the protein levels of Parkin were not only reduced in BAT, but also in eWAT and iWAT, which indicates that it’s not a brown adipocyte-specific model.

We have changed the subtitle to be more specific “Loss of Parkin in BAT fails to improve glucose homeostasis.”

We have added the information about the generation of the brown-adipose tissue-specific Parkin knockout mouse model into the method section. Validation blots are updated in Suppl. Figure 3C.

7. In the Parkin^{BAT} mice, is there any difference in body weight after HFD feeding? Which time point was used for the glucose tolerance test in Figure S3G?

Yes, Parkin^{BAT} mice have less body weight gain than control mice during HFD feeding. However, the blood glucose levels during GTT analysis were not altered between groups with either a normal chow diet or HFD feeding.

To elucidate the early metabolic response to HFD feeding, we performed GTT on Parkin^{BAT} mice after 5 weeks of HFD feeding (the same time we performed GTT on Parkin^{Adi} mice). We have added this information to the figure legend of Suppl. Figure 3G.

8. Does Parkin KO also promote mitochondrial biogenesis in iWAT and BAT? If not, why is the mechanism specific to eWAT?

Thank you for your question. Indeed, Parkin KO increases mtDNA content in BAT and enhances browning in iWAT. We are investigating BAT-specific mechanisms, which we hope to publish in a follow-up manuscript.

9. The body weight changes for HFD-fed control and Parkin^{Adi} in Figure S4A were not consistent with those in Figure 2A.

The mice we studied in Figure 2A and Figure S4A are two different cohorts of mice. To minimize the impact of body mass on metabolic rate (Nat Metab, 2021, 3(9): 1134-1136. PMID: 34489606.), we selected weight-matched mice to measure energy expenditure. We do not deny that there are minor differences in body weight between cohorts. Here we provided the body weight and tissue weights of HFD-fed Control f/f and Parkin^{Adi} mice (on the right) on which we performed indirect calorimetry studies. Although the body weight was not significantly altered between groups at 4 wks, we observed lower body weight in the following wks of HFD feeding and less white adipose tissue mass at week 8. These results are consistent with our previous findings in Figures 2A and 2B.

Minor concerns:

1. Please provide the age of mice in Figure S2Q.

The age of these mice has been added to the figure legend.

2. For the IP experiment in Figure 5F, there are bands in the IgG group Figure 5F.

The association of Nqo1 protein with Pgc1 α has been detected *in vitro* by Dr. Yousef Shaul's group (Mol Cell Biol, 2013, 33(13):2603-13. PMID: 23648480.). Here, we validated this association in eWAT lysate by Co-IP. We did realize there is a band in the IgG group, but that band is above 100 kDa, higher than the molecular weight of Pgc1 α (91kDa).

Reviewer #2 (Remarks to the Author):

Moore et al. took advantage of mice on high-fat diet with genetic intervention of Parkin in adipose tissue to elucidate the mechanism by which targeting Parkin in adipocytes prevents obesity by regulating mitochondrial homeostasis through mitophagy and mitochondrial biogenesis. This is clearly one of the most important area with great potential impacts on obesity and metabolic syndromes. The concept is also novel. However, this reviewer has identified many problems with the data ranging from the experimental design, validity of the findings and the quality of the data. Overall, the findings do not warrant the conclusion that Parkin regulates mitochondrial biogenesis and mitophagy through the regulation of Pgc-1 α protein stability via Nqo1.

1. Data presented in Fig. 1 showed that Park2 (Parkin) mRNA increases during adipogenic differentiation in both immortalized cells lines and primary adipocytes as well as in eWAT in mice. These findings establish the potential correlation of Parkin expression with adipose tissue expansion, which is the foundation of this paper. The identity of the band shown as Parkin needs validation. If the bands in eWAT and iWAT (c and d) are true Parkin bands, the expression in SVF should be interpreted as not detectable rather as an arbitrary unit of 1.

We thank reviewer 2 for the comments. We have repeated the blots and updated Figures 1C and 1D.

2. Fig. 1E is confusing. 0 day was described as SVF fractions from eWAT and 10 day as 10 day-differentiated adipocytes, why it is labeled as eWAT-SVF. The quantification of this panel is not trustworthy.

We repeated the experiments and updated the blot for Figure 1E. We isolated SVF from eWAT and differentiated them into adipocytes. 0, 6, and 10-day-differentiated primary adipocytes from eWAT SVF fractions.

3. Fig. 2 lacks control. Control and Parkin^{Adi} on normal chow diet must be included. GTT and ITT are relatively crude assays for whole body metabolism. This reviewer has not previously seen such tight data from mice with a sample size of 6-8 as presented in Fig. 2C and 2D. Appropriate controls (groups on NC), the GTT and ITT data should be included, which will be revealing. Fig. 2E should be quantified for each mouse. Fig. 2G is difficult to evaluate. Could not see the image. No quantification.

The data of normal chow diet-fed Control and Parkin^{Adi} mice were provided in Suppl. Figure 2M to Q. We have marked the time for GTT and ITT in Suppl. Figure 2M. The reasons we did not combine NC and HFD-fed GTT together are (1) we fasted NC-fed mice for 6 hours before GTT while HFD-fed mice were fasted overnight, causing different basal glucose levels; (2) NC and HFD-fed mice are different cohorts and we did not perform GTT at the same time, which could increase variability between groups.

We have added the quantification for Figures 2E and 2G.

4. Same issue for NC groups for Fig. 3. Fig. 3I showed data from mice on NC. Why not include all data in Fig. 1 and Fig. 2?

NC-fed Parkin^{Adi} mice do not display phenotype before 9 months of age, but these mice also have increased mitochondrial DNA encoded genes and mtDNA content in eWAT, like HFD-fed Parkin^{Adi} mice. These findings strengthen our statement that Parkin deletion in WAT elevates mtDNA content. Therefore, we only combined the molecular study data from both NC and HFD-fed mice in Figures 3I and 3J.

5. The data in Fig. 3I suggest that gene deletion of Parkin in adipocytes leads to enhanced gene expression related to mitochondrial biogenesis under NC condition. Fig. 3A-3E showed increased VO₂, VCO₂, and EE with slightly decreased RER in Parkin AKO mice. It is critical to show the locomotive activity during the dark and light cycle. If Parkin AKO mice on HFD have significantly higher cage activities, it could potentially explain all the findings presented in this paper.

We have updated Suppl. Figure 4I with the locomotive activity during the light and dark cycle. There is no change in locomotive activity. Therefore, we believe the lean phenotype is not a secondary phenotype of locomotor activity.

6. Most importantly, Parkin AKO needs to be validated. No data show how effective the deletion of the Park2 gene in adipocytes or WAT/BAT in Parkin AKO mice. The same is true for ParkinKD adipocytes.

We repeated the validation of our models (Parkin^{Adi} and Parkin^{BAT}). New data have been included in Suppl. Figure 2E and Suppl. Figure 3C. Results of Parkin knockdown validation in adipocytes are provided in Suppl. Figure 4O.

7. The statistical analysis for Fig. 3M is problematic. Each individual cell cannot be counted as a sample. The mean value of the measured cells in the same well is considered as n = 1. The same issue is also raised for other figures in this manuscript.

We have increased the sample size and updated all the figures where necessary.

8. The data presented in Fig. 4 has the same issue of lacking NC control and the lack of evidence of adipose-specific KO of Parkin.

Since HFD-fed Parkin^{Adi} mice show phenotype much earlier and more severe than NC fed-mice, we decided to determine the protein levels of autophagy markers and mitochondrial biogenesis only in HFD-fed mouse models. The results of Parkin KO validation are updated in Suppl. Figures 2E and 3C.

9. The interpretation of reduced p62 as evidence of reduced autophagy is not warranted. The level of p62 protein is an indication of the level of one of the cargo proteins for autophagy, but itself is degraded during the process of autophagy. With no other supporting evidence, it is premature to interpret the data as evidence of decreased autophagy.

We fully agree with the reviewer. We have altered conclusions in the paper to reflect this view.

10. Fig. 4C-4D has the same problem of using the number of cells from the same well as the number of samples. The use of the construct of GFP-mCherry-Fis1 for quantification of mitophagy needs to be validated. The data as presented is not as convincing to suggested impaired mitophagy by Parkin KD.

We received a generous gift of adenovirus GFP-mCherry-Fis1 from Dr. Orian Shirihai's laboratory at UCLA. The construct has been validated by his group (Biochem J, 2020, 477(2): 461-475. PMID: 32003437; Nat Commun, 2020, 11(1):3347. PMID:32620768). We have increased the sample size to strengthen our findings and updated the Figures.

11. The image of Fig. 4F is cropped. The same size (n=3) is small. Cropping image data needs to be explained.

We thank reviewer 2 for the concern. The images were not cropped. All immunoblotting data in Figure 4F were captured from the same membrane. A stripe in the 2nd band of Pgc1a was generated during the imaging not by cropping.

Raw images are provided. We also provide another set of Pgc1a blots (n=3, on the right). Moreover, Pgc1a protein level is consistently higher in Parkin^{KD} adipocytes at each time point in Figure 4G.

The shrank OXPHOS protein panel (Atp5a, Uqcrc2, mtCo1, Sdhb, and Ndfub8) caused the size mismatch. This happened when we organized the image panel in the GraphPad Prism software. We have corrected the size of the OXPHOS panel.

12. The identity of Pgc-1a protein needs to be validated as it is the focus of the study. Pgc-1a is a very labile protein with a half-life in the range of 1-2 h (Rasbach et al. Arch Biochem Biophys PMID 18718443, 2008; Park et al. PNAS, 2015 PMID 25918410). CHX treatment blocks translation, which should always result in reduced protein abundance over time. It is strange that under both control and KD conditions, Pgc-1a protein increased at least in the first 4 hours.

We thank reviewer 2 for the comments. We found that in Rasbach et al. Arch Biochem Biophys PMID: 18718443, 2008- the authors studied renal proximal tubule cells in this paper, and in Park et al. PNAS, 2015 PMID: 2591841- the authors studied HEK293 cells in this paper. We believe that Pgc1a protein stability is different between cells. This has been proved by many papers (PNAS, 2010, 107: 14508-14513. PMID: 20699386; Nat Commun, 2018, 9(1):689. PMID: 29449567). Moreover, we repeated the CHX studies 3 independent times. We are confident in our findings.

13. It is also peculiar that CHX treatment causes increased Nqo1 for 8 hours in Fig. 5E. The data presented in Fig. 5H needs to be quantified with a sufficient sample size.

We performed CHX studies 3 times for each time point. The densitometry analysis is based on these 3 independent studies.

We performed 2 more independent Co-IPs of Pgc1a and added the quantification result in Figure 5H. Co-IP results were added in Suppl. Figure 5C.

14. The data presented in Fig. 6A does not make logical sense. HFD caused an increased Parkin expression (Fig. 1F), which impaired mitochondrial gene expression (Fig. 3G) and repressed Pgc-1alpha gene expression (Fig. 3I). ParkinAdi mice had increased Nqo1 (Fig. 5A), suggesting that Parkin inhibit Nqo1 expression and/or promote Nqo1 degradation. If anything, Ob/+ mice (more like HFD mice) should have reduced Nqo1 expression, which is the opposite at least for Fig. 6A. A critical piece of data is to show that in the absence of Nqo1 protein induction, Pgc-1alpha protein induction under the condition of Parkin KO/KD is gone to establish the causal relationship although it does not negative the value of the findings for enhanced Nqo1 expression as a therapeutic potential of treating obesity.

We thank the reviewer for this comment. We agree that this is a complex process. We proposed that Parkin deletion in WAT elevates mtROS, which increases Nqo1 protein level. Our mitoQ and Menadione studies suggest that mtROS is the key to mediating Nqo1 expression (Figure 5J, 5K, and 5L). However, we do not have enough evidence to claim that “Parkin inhibits Nqo1 expression and/or promotes Nqo1 degradation” mentioned by Reviewer 2. This idea is not our focus and it needs to be further interrogated.

We found Nqo1 gene expression positively associates with adiposity (Figures S6A and S7A-B). However, other studies found that Nqo1 transgenic mice could prevent diet-induced obesity (NPJ Aging Mech Dis, 2020, 6(1):13. PMID: 33298924.). These conflicting findings suggest that Nqo1 gene expression may not be concordant with its protein level. To prove this hypothesis, we examined both gene expression and protein level of Nqo1 in Ob/+ mice (a genetic mouse model with a heterozygous leptin mutation), because Ob/+ mice have no change in Parkin gene expression and protein level (protein data was not shown, trying to exclude the impacts of Parkin on Nqo1). As reviewer 2 mentioned, this model is not equivalent to the mice with HFD feeding. Indeed, we found that Nqo1 gene expression, but not protein level, was elevated in Ob/+ mice (Figure 6A). The reason we presented the inconsistent results of Nqo1 mRNA expression and protein level is to show that Nqo1 gene expression-related association without the protein level and/or enzymatic activity determination is sometimes not enough to make a conclusion.

Parkin and Nqo1 double KD results were provided in Suppl. Figure 5B. We found double KD Parkin and Nqo1 decreased the protein level of Pgc1a in 3T3-L1 adipocytes. Moreover, both Nqo1 inhibitors Dicoumarol and ES936 significantly suppressed the protein level of Pgc1a in Parkin^{KD} adipocytes (Figures 5G and S5A). Together, these findings suggest that an elevated Nqo1 level is a trigger to increase Pgc1a and mitochondrial biogenesis in adipocytes.

REVIEWERS' COMMENTS

Reviewer #1 (Remarks to the Author):

The authors had sufficiently addressed the concerns and I have no further comments.

Reviewer #3 (Remarks to the Author):

The authors have addressed the comments of reviewer #2 properly. The authors observed that Parkin deficiency leads to the enhanced PGC1 stability, thereby the increased mitochondrial biogenesis via Nqo1. They further suggest that Parkin regulates mitochondrial homeostasis by balancing mitophagy and Pgc1 α - mediated mitochondrial biogenesis in white adipocytes. This is an interesting topic. Previous studies have shown that Parkin may regulate PGC1 activity via PARIS. Knockout of FUNDC1 or BNIP3 also affects mitochondrial biogenesis. They should include these work in their discussion and discuss the complexity of the mitochondrial homeostasis.

Again, we thank all reviewers for the efforts taken to review our manuscript.

Reviewer #3 (Remarks to the Author):

The authors have addressed the comments of reviewer #2 properly. The authors observed that Parkin deficiency leads to the enhanced PGC1 stability, thereby the increased mitochondrial biogenesis via Nqo1. They further suggest that Parkin regulates mitochondrial homeostasis by balancing mitophagy and Pgc1 α - mediated mitochondrial biogenesis in white adipocytes. This is an interesting topic. Previous studies have shown that Parkin may regulate PGC1 activity via PARIS. Knockout of FUNDC1 or BNIP3 also affects mitochondrial biogenesis. They should include these work in their discussion and discuss the complexity of the mitochondrial homeostasis.

Thank you for your suggestions. We have included these in the discussion.